# Natural Polymers in Heart Valve Tissue Engineering: Strategies, Advances and Challenges

**DOI:** 10.3390/biomedicines10051095

**Published:** 2022-05-08

**Authors:** Diana Elena Ciolacu, Raluca Nicu, Florin Ciolacu

**Affiliations:** 1Department of Natural Polymers, Bioactive and Biocompatible Materials, “Petru Poni” Institute of Macromolecular Chemistry, 700487 Iasi, Romania; nicu.raluca@icmpp.ro; 2Department of Natural and Synthetic Polymers, “Gheorghe Asachi” Technical University of Iasi, 700050 Iasi, Romania

**Keywords:** heart valve tissue engineering, polysaccharides, proteins, scaffold, heart valve replacement, regenerative medicine

## Abstract

In the history of biomedicine and biomedical devices, heart valve manufacturing techniques have undergone a spectacular evolution. However, important limitations in the development and use of these devices are known and heart valve tissue engineering has proven to be the solution to the problems faced by mechanical and prosthetic valves. The new generation of heart valves developed by tissue engineering has the ability to repair, reshape and regenerate cardiac tissue. Achieving a sustainable and functional tissue-engineered heart valve (TEHV) requires deep understanding of the complex interactions that occur among valve cells, the extracellular matrix (ECM) and the mechanical environment. Starting from this idea, the review presents a comprehensive overview related not only to the structural components of the heart valve, such as cells sources, potential materials and scaffolds fabrication, but also to the advances in the development of heart valve replacements. The focus of the review is on the recent achievements concerning the utilization of natural polymers (polysaccharides and proteins) in TEHV; thus, their extensive presentation is provided. In addition, the technological progresses in heart valve tissue engineering (HVTE) are shown, with several inherent challenges and limitations. The available strategies to design, validate and remodel heart valves are discussed in depth by a comparative analysis of in vitro, in vivo (pre-clinical models) and in situ (clinical translation) tissue engineering studies.

## 1. Introduction

*Cardiovascular diseases* are the leading cause of death globally, and among them, aortic stenosis, determined by the calcification of a trileaflet valve (degenerative calcific aortic valve stenosis, CAVS) or stenosis of a congenital bicuspid valve (congenital bicuspid aortic valve, CBAV), is the most prevalent form of cardiovascular disease in the world, after hypertension and coronary artery disease [1,2,3].

The common treatment for heart valve disease is surgical replacement, but because of the lack of organ donors, alternative approaches are essential for restoring the cardiac function after a heart attack. Surgical replacement of diseased heart valves has been widely performed, primarily with mechanical valves and bioprosthetic heart valves. All these devices have significant limitations with risks of further morbidity and mortality: mechanical valves may cause hemorrhage and thromboembolism, and thus, they require lifelong anticoagulation treatment; bioprosthetic valves have relatively poor long-term durability because of degeneration, calcification and fibrosis, and may cause immunogenic complications [4,5,6].

These difficulties have motivated the development of *tissue engineering strategies* for valve substitution, which are intended to achieve valve replacements that are based on a three-dimensional (3D) structure capable of supporting cell proliferation, differentiation and growth (in vitro or in vivo) in a functional tissue construct [7]. The main function of *heart valves* (HV) is to maintain unidirectional blood flow during cardiac systole and diastole, knowing that the normal heart valves open and close about 4 million times a year without obstruction or regurgitation [8,9]. Thus, *heart valve tissue engineering* (HVTE) requires complex substrate geometries to provide for optimal opening and closing behavior of the valve leaflets [10].

Over the past few decades, several studies have been performed to clarify the desirable characteristics of *tissue-engineered heart valves* (TEHV) and to develop strategies for generating these valve substitutes [11,12,13,14].

This review provides a synthesis of the HVTE studies, emphasizing the principles, the recent advancements, the current challenges and the future directions in this field. Starting from the basic principles of tissue engineering, the advantages and limitations of the scaffolds are emphasized. Different techniques for heart valve replacements fabrication, as well as the evolution of in vitro, in vivo and in situ strategies for tissue engineering applications are also discussed. The complex and dynamic structural components that are needed to accomplish normal heart valve function and the required steps to design and validate novel valves are described, particularly focusing on the natural polymers used in recent years in heart valve tissue engineering.

## 2. Scaffolds for Tissue Engineering: General Concepts

***Tissue engineering*** is a rapidly advancing field in regenerative medicine, with various research papers directed toward the production of new biomaterial scaffolds with tailored properties which can be used to restore, maintain or improve damaged tissues or even whole organs [15]. A key concept in tissue engineering is to restore and improve the function of the tissues by preparing porous three-dimensional scaffolds, and seeding them with cells and growth factors. These three things, i.e., scaffolds, cells and growth factors, are known as “the tissue-engineering triad” and this system is set up in an appropriate environment in a bioreactor [16].

One of the most important entities to be considered for efficient tissue engineering is the ***scaffold***, because its external geometry, surface properties, pore density and size, interface adherence, biocompatibility, degradation and mechanical properties affect not only the generation of the tissue construct in vitro, but also its post-implantation viability and functionality [17,18].

Multiple scaffolds have been designed, developed and tested, and thus, nowadays, different ***types of scaffolds*** are available in the field of tissue engineering. In general, these can be classified into two main groups: (i) *acellular scaffolds*, such as decellularized human or animal tissue, and (ii) *artificial scaffolds*, fabricated from natural or synthetic polymers and composites.

***Acellular scaffolds*** are the ideal bio-scaffolds necessary to guide host or donor cells toward the regeneration of new and functional tissues and are obtained upon the removal of nuclear content and cellular elements, the scaffolds retaining the architecture and complexity of the native tissues, including vasculature and bio-factors present in the extracellular matrix (ECM) [19]. The obtained acellular or decellularized matrices slowly degrade upon implantation and are generally replaced by the ECM proteins secreted by the in growing cells. The advantages of these scaffolds lie in the removal of all foreign cells and immunogenic compounds, and the retention of their correct anatomical structure and the similar bio-mechanical properties to those of native tissues (such as signaling for cell adhesion and induction of cell migration, proliferation and differentiation), which are critical for the long-term functionality of the grafts [20]. Acellular tissues are biocompatible and the absence of rejection after allogeneic or xenogeneic transplantation makes them the ideal scaffolds for translational medicine applications and organ replacement [21,22]. The obvious advantage of this scaffold is that it is composed of ECM proteins typically found in the body. Naturally derived materials and acellular tissue matrices have the potential advantage of biological recognition. Polymer coating of a tissue-derived acellular scaffold can improve the mechanical stability and enhance the hemocompatibility of the protein matrix.

The decellularization process consists of removing the cellular material from the ECM of biological tissues, leading to a semiporous scaffold (remaining ECM), minimizing damage to the original structure and maintaining the same complex geometry of the native tissue. The scaffold obtained contains natural components (collagen, elastin and glycosaminoglycans) that provide clues for cell migration and differentiation, resulting in a constructive remodeling.

*Decellularized heart valves* have been more clinically relevant than polymeric valves, due to (i) their positive answer regarding cell differentiation (natural components that can positively impact cell differentiation), (ii) the remodeling process, when these serve as building blocks, (iii) maintaining the mechanical anisotropy of the native valves and, furthermore, (iv) they do not necessitate complete biodegradation. However, decellularized heart valves require human or animal tissue for manufacture, which is limited in supply, and necessitates cryopreservation for storage. The successful use of decellularized heart valves depends on the decellularization process and on the immune response following implantation. The freeze-drying method of biologic heart valves has been used to facilitate long-term storage. Unfortunately, certain limitations of this method have been found, specifically, the collapse of the ECM structure and disruption of biomolecules during the freeze-drying process. To overcome these limitations, the use of lycoprotectants has been proposed [23].

The search for alternative solutions to replace acellular scaffolds leads the research toward the scaffolds fabricated from polymeric materials, which can be categorized into porous, microsphere, hydrogel and fibrous scaffolds.

***Porous scaffolds*** are a 3D structure with an interconnected homogeneous pore network, providing a continuous flow of nutrients and metabolic waste to enable growth and vascularization of engineered tissues. Porous scaffolds can be manufactured using biopolymers with a specific surface-area-to-volume ratio, crystallinity, pore size and porosity [20]. The preparation techniques can be divided into two categories: (i) *non-designed manufacturing techniques*, which include freeze drying or emulsion freezing, melt molding, phase separation, solvent casting or particulate leaching, gas foaming or high-pressure processing, electrospinning and combinations of these techniques, and (ii) *designed manufacturing techniques*, which includes rapid prototyping and 3D printing [24]. Generally, conventional fabrication techniques do not enable precise control of internal scaffold architecture (pore size, pore geometry, pore interconnectivity, spatial distribution of pores and construction of internal channels within the scaffold) or the fabrication of complex architectures that could be achieved by rapid prototyping techniques, for example [25]. Rapid prototyping (RP), generally known as solid free-form fabrication or additive manufacturing, is a group of advanced manufacturing processes in which objects can be built layer by layer in additive manner directly from computer data, such as computer-aided design (CAD), computed tomography (CT) and magnetic resonance imaging (MRI) data [26]. Recently, 3D printing has emerged as a promising technology for fabricating geometrically defined porous architectures in 3D, thereby efficiently improving the physiological relevance of tissues and overcoming the significant limitations of various scaffold-based approaches [27]. Regarding the design of 3D printed porous scaffolds that simulate tissues, some properties to keep in mind are: surface area and interconnectivity, which are related to cell growth; permeability, which governs nutrient transport; and mechanical strength, which assures support and protection, among other properties [28]. The most commonly used approaches in developing 3D printed models include selective laser sintering (SLS), fused deposition modeling (FDM), inkjet printing (IJP), multi-jet modeling (MJM), extrusion-based approach and laser-based stereolithography (SL) [29,30]. Porous scaffolds exist in different forms, such as sponge, foam, mesh and nano- and microscale biodegradable fibers; the last two types can indeed be categorized under fibrous scaffolds [31]. Within this category of scaffolds, sponge or foam porous scaffolds have been used in tissue engineering applications [20].

An *ideal porous scaffold in heart valve tissue engineering* should exhibit a native extracellular matrix (ECM) texture to support repair and regeneration processes. The tissue-engineered valve scaffolds obtained by the conventional techniques, such as particulate leaching, solvent casting, gas foaming, vacuum drying, thermally induced phase separation, melt molding, high internal phase emulsion and microfabrication [32,33], have pores with irregular sizes, which are not interconnected, and more importantly, lack features such as shape and elastomeric flexibility. Recently, in order to create anatomic models, the scaffolds have been prepared by using computer-controlled tools for layer-by-layer deposition of materials or 3D printing [34]. With the advancement of 3D printing technique, a heterogeneous 3D scaffold with strong mechanical strength and with all required characteristics of an ideal scaffold for cardiac tissue engineering, such as the morphology and accuracy of native ECM, can be fabricated. In order to develop the scaffolds intended for heart valve engineering, a bioink composed of cells and desired biomaterials is used to print the specific shape of the organ. Three-dimensional printing-based applications of tissue engineering in combination with stem cell technology have the potential to address the shortage of donor organs for transplantation and provide patient-specific tissue replacement [35].

***Microsphere scaffolds*** are increasingly used as drug delivery systems and in advanced tissue engineering applications such as gene therapy, antibiotic treatment of infected bone and so forth [36]. Regarding the methods used to fabricate microspheres, these are the emulsion-solvent extraction method, precision particle fabrication (PPF) and thermally induced phase separation (TIPS), while the methods used to produce microsphere-based scaffolds as a single macroscopic unit are: (i) heat sintering, (ii) solvent-based sintering (solvent vapor sintering and weak solvent sintering), (iii) subcritical CO_2_ sintering and (iv) selective laser sintering (SLS) [37]. Microspheres as building blocks have various benefits, such as simple method of preparation, controlled morphology and physico-chemical characteristics and controlled release of encapsulated factors [20]. Densely packed microsphere-based porous scaffolds can both serve as a template for cell proliferation and act as a guide for establishing intricate cell–cell/cell–ECM connections, which permits their utilization in regenerative engineering.

In *cardiac tissue engineering*, an important challenge is the design of myocardium, which must be highly porous to allow the nutrients’ passage to the cells and to enable formation of aligned and electrically interconnected cardiomyocytes. The spherical nature of microspheres permits a dense packing in regular arrangements, which can be tailored to meet the specific tissue requirements [37].

***Hydrogel scaffolds***. Over the past decades, an increasing demand for scaffolds to guide the growth of new tissues has led to the development of new strategies for the production of hydrogels with applications in the revolutionary field of tissue engineering. These can be prepared from synthetic or natural polymers, which are physically cross-linked (reversible) or chemically cross-linked (irreversible), and the cross-linking bonds could be covalent or non-covalent (hydrogen bonds, ionic or hydrophobic interactions) [38,39,40]. Hydrogels based on natural polymers have various advantages, such as biocompatibility, cell-controlled degradability and intrinsic cellular interaction, while synthetic polymer-based hydrogels can be prepared with precisely controlled structures and functions [20]. In addition, the combination of natural and synthetic polymers can be used to provide proper scaffold degradation behavior after implantation. Hydrogels are considered biocompatible, due to the structural similarity to the ECM found in tissues, and need specific requirements to function appropriately and promote new tissue formation. These requirements include both physical parameters (in vivo swelling properties, mechanical strength, biodegradation properties), as well as biological performance parameters (cell adhesion and proliferation). Their compatibility with biological tissues, high water content and good mechanical properties make hydrogels particularly attractive for tissue-engineering applications. By adding cells to a hydrogel before the gelling process, these can be distributed homogeneously throughout the resulting scaffold. Fibroblasts, osteoblasts, vascular smooth muscle cells and chondrocytes successfully immobilize and attach to these hydrogel scaffolds [41].

Tissue engineering techniques used three types of hydrogels for cardiac tissue engineering, and those are: (i) *natural polymer-based hydrogels*, materials derived from a biological source, either animals, plants or algae, such as collagen (COL), fibrin (F), hyaluronic acid (HA), alginate (Alg), gelatin (Gel), chitosan (CH), etc.; (ii) *synthetic polymer-based hydrogels*, such as poly(ethylene glycol) (PEG), poly(ethylene glycol) diacrylate (PEG-DA), polycaprolactone (PCL), polylactic acid (PLA), poly(lactic-co-glycolic acid) (PLGA), polyacrylamide (PAM), polyurethane (PU), etc.; and (iii) *composite hydrogels*, which combine the advantages of both synthetic and natural polymers [18,42,43,44,45]. These materials are used to fabricate hydrogel scaffolds that mimic the native ECM and present similar morphology.

***Fibrous scaffolds*** are superior scaffolds in terms of cell adhesion, migration, proliferation and differentiation, due to the high aspect ratio of fibers, growth factor loading efficiency and sustained release capacity. Different techniques are available for preparation of nanofibrous materials, such as electrospinning, self-assembly, phase separation, jet-spraying, jet-spinning, double component electrodeposition and, more recently, melt electro-writing [46,47,48,49,50]. Among these, electrospinning is the most widely used technique and with the most promising results for tissue engineering applications, due to easy handling, applicability to most polymers and cost-effectiveness. The development of nanofibers has enhanced the scope for fabricating scaffolds that can potentially mimic the architecture of natural human tissue at the nanometer scale.

For *heart valve tissue engineering,* fibrous scaffolds would provide an ideal environment for cells, if they could form 3D structures with porosity, pore size and mechanical characteristics comparable to native heart valves. Various polymers have been used for HVTE, such as polyglycolic acid (PGA), PLGA, PLA, poly L-lactic acid (PLLA), PCL, poly(L-lactic acid-co-ε-caprolactone) (PLCL) and PU as synthetic polymers and COL, Gel, CH and HA as natural polymers [18,49,50,51,52].

Regardless of the ***scaffold specific properties***, a number of key considerations are important when designing or determining the suitability of a scaffold for use in tissue engineering, as described below.

The ***porous architecture*** of scaffolds used for tissue engineering should have an interconnected pore structure and adequate mean pore sizes, large enough to ensure cellular penetration and small enough to establish a sufficiently high specific surface [25]. If pores are too small, cell migration is limited, resulting in the formation of a cellular capsule around the edges of the scaffold, which can limit the diffusion of nutrients and the removal of waste, resulting in necrotic regions within the construct [53]. If pores are too large, limited cell adhesion was observed due to a decrease in surface area. Therefore, the critical dimension of pores may vary depending on the cell type used and the tissue being engineered. In addition, the scaffold must allow an adequate diffusion of nutrients to cells and the ECM formed by these cells, as well as the diffusion of waste products out of the scaffold [54].

The produced scaffold should have adequate ***mechanical properties***, to mimic the anatomical site where it is intended to be implanted, and to function from the time of implantation to the completion of the remodeling process [55]. A scaffold’s mechanical properties (strength, modulus, toughness and ductility) are determined both by the material properties of the bulk material and by its structure (macrostructure, microstructure and nanostructure). Matching the mechanical properties of a scaffold to the graft is critically important, so that the progression of tissue healing is not limited by its mechanical failure prior to complete tissue regeneration [56]. Many materials have been produced with good mechanical properties, but to the detriment of retaining high porosity. In addition, many of these materials, with demonstrated in vitro potential, have failed when they were implanted in vivo because of insufficient capacity of vascularization [53]. Thus, to achieve a suitable scaffold, it is necessary to balance the mechanical properties with a porous structure, sufficient to allow cell infiltration and vascularization.***Interface adherence*** of the scaffold referred to the interactions between cells and their environment, which play a critical role in determining cell fate and physiological functions, so as to maintain normal phenotypic shape within the scaffold. An ideal scaffold should provide informative microenvironments mimicking physiological niches to direct advanced cell behaviors, such as differentiation, proliferation and apoptosis, without inducing pathological outcomes, such as calcification [4].The scaffold’s ***biocompatibility*** is related to the cell’s adherence, which should function normally, migrate onto the surface or even through the scaffold, begin to proliferate and, finally, have a negligible immune reaction. Thus, to be accepted in vivo, the host immune response should be minimal for the scaffold. The *biocompatibility of the cross-linking agent* used is particularly important, especially in cases where reactive groups of the cross-linker are incorporated into the hydrogel network and might then be released upon degradation. Although unreacted chemicals are usually eliminated after cross-linking through extensive washing in distilled water, as a rule, toxic cross-linkers should be avoided, in order to preserve the biocompatibility of the final scaffold [53].A scaffold should be ***biodegradable*** and the degradation products should be non-toxic and able to be eliminated from the body without interference with other organs. There are different mechanisms for in vivo degradation, such as hydrolysis, oxidation, enzymatic and physical degradation [57]. The biodegradation process permits to the cells to produce their own extracellular matrix and finally to replace the implanted or tissue-engineered constructed scaffolds, eliminating the need for further surgery to remove it. The scaffold’s degradation rate should be adjusted to match the rate of tissue regeneration so that it has disappeared completely once the tissue is repaired [58,59].

The advantages and disadvantages of the above presented scaffolds, such as porous, microsphere, hydrogel and fibrous scaffolds, are summarized in Table 1 [18,20,37].

***Scaffolds intended for heart valve tissue engineering*** face additional distinct challenges owing to their direct contact with blood. Specifically, the construct should be resistant to calcification, should have a *minimal thromboembolism risk* and must *withstand the unique hemodynamic pressures and flows of the cardiac environment* from the moment of implantation [18,35]. Moreover, the scaffold should imitate the natural myocardial ECM and should possess adequate porosity that promotes vascularization (Figure 1).

It should also allow continuous diffusion of oxygen and nutrients to the seeded cells and it must mimic the mechanical properties of the native cardiac tissue and bear the cyclic strains and stresses exerted upon transplantation, and must also be sufficiently thick to contract with proper strength and beat synchronously with the neighboring cardiomyocytes [35,57].

These unique challenges underline the importance of carefully considering the materials and design when fabricating a scaffold for tissue-engineered heart valves.

***Ideal heart valve tissue-engineered scaffolds*** are defined as three-dimension porous solid biomaterials designed to perform some or all of the following functions: (i) promote cell-biomaterial interactions, cell adhesion and ECM deposition, (ii) permit sufficient transport of gases, nutrients and regulatory factors to allow cell survival, proliferation and differentiation, (iii) biodegrade at a controllable rate that approximates the rate of tissue regeneration under the culture conditions of interest and (iv) provoke a minimal degree of inflammation or toxicity in vivo [20]. Scaffolds can be seeded with embryonic or adult stem cells, progenitor cells, mature differentiated cells or co-cultures of cells to induce tissue formation in vitro and in vivo. While the specific functions vary with tissue type and clinical need, scaffolds may potentially coordinate biological events at the molecular, cellular and tissue levels on time and length scales ranging from seconds to weeks and nanometers to centimeters, respectively. A central theme in designing tissue-engineered scaffolds is to understand the correlations between scaffold properties and biological functions [60].

## 3. Heart Valve Replacements

The heart contains four chambers (two atria and two ventricles) and four valves (Figure 2): (i) the tricuspid valve, serving as blood flow regulator, from the left atrium to the ventricles, (ii) the pulmonary valve, which controls the blood flow from the right ventricle to the pulmonary artery, (iii) the mitral valve, which regulates the blood inflow from the right atrium to the ventricles, and (iv) the aortic valve, having the role of regulating the flow from the left ventricle to the aorta [61].

This review dwells on two valves of the four (aortic valve and pulmonary valve), with a major focus on the aortic valve. Both these valves have similar structures and mechanical characteristics, the differences appearing in the thickness of the layer and its density. As a structure, they consist of three semicircular leaflets (cusps), which are connected to a fibrous annulus (root).

The most common valve disease is the aortic valve stenosis (determined by the calcification and thickening of the cups), which is presented concomitantly with aortic regurgitation (because of the loss of stretch in calcified cups) [63]. Generally, the valve dysfunction is caused by either aging or congenital defects, and the severe complications of these degenerative diseases can seriously affect the structure and function of heart valves. In the short term, medication may be used to improve the health of patients, but for patients with severe valvular pathologies, the best option is to do surgery in order to repair or even replace the valve.

It is well known that the first step toward heart valve replacements is to ensure long-term functionality of implantation and a competent and stable structure with specific anatomical and histological features [64,65].

There are three categories of heart valve replacements, of which the most two common categories are the mechanical and bioprosthetic valves [4]. The development of the polymeric valves was intended to overcome the problems characteristic of the above-mentioned valves, related to regeneration, growth potential and durability.

The classification of heart valve replacements, as well as their main advantages and disadvantages, are presented in Table 2.

### 3.1. Mechanical Valves

Mechanical valves were developed in a diversity of shapes and sizes (caged-ball, tilting disk and bi-leaflet), due to their high durability (more than 20 years) and better fluid mechanics (Figure 3) [4,66,67,68].

The first mechanical valve, the *caged ball valve* (Hufnagel valve), consisted of a methacrylate chamber containing a methyl methacrylate ball that was placed in the descending thoracic aorta instead of the heart itself, with very poor hemodynamic performances. The next improved model consisted of a methacrylate cage and a silicone elastomer rubber-ball (Starr-Edwards valve), with the first promising results. However, because of different complications associated with these types of valves, the *non-tilting disc valves* were developed, fabricated from a silicone elastomer disc and a stellite housing (Kay-Shiley valves) [63]. In order to avoid the health problems induced by silicone, the next model replaced it with a Delrin polymer disc with a markedly improved durability. The *tilting disc valves* (mono-leaflet valves) developed from Delrin polymer (Bjork-Shiley flat disc valve) and later replaced by a pyrolyte disc (Medtronic-Hall tilting disc valves) was extremely successful worldwide. A major problem of this type of valve was the fracture of the weld site of the small C-shaped outflow of strut [66], which led to the development of the last generation of mechanical valves, the *bileaflet valves*. Thus, the evolution of mechanical valves, corroborated with the discoveries from the surgical processes, allowed the realization of a new generation of valves, which present physiological geometries and improved hemodynamic characteristics. Today, bileaflet valves have become the most widely implanted valves and pyrolyte is the most used biomaterial for production of the inner orifice [63,69].

However, in addition to all these benefits, there are significant risks that may arise after implantation, owing to the reaction of the human immune system to the introduction of foreign materials into the body, such as thrombosis and infections. Thrombosis occurs because of the adsorption of blood proteins such as fibronectin, vitronectin, fibrinogen and von Willebrand factor, onto the scaffold surface, causing blood contact activation, platelet activation and thrombin and fibrin formation in blood plasma [18,70].

In order to avoid these problems, the patients must receive anticoagulant drug therapies for the rest of their lives, a consequence that involves inevitable risks of hemorrhagic complications. Thus, these valves are contraindicated to the athletes, who make a sustained effort, and also to young women, because the anticoagulation therapy can lead to abnormal fetus development and to increased risks of birth-related bleeding [65,71].

### 3.2. Bioprosthetic Valves

*Bioprosthetic valves* can be classified in *autografts* and *allografts* (homografts), derived from humans, and *xenografts* (heterografts), derived from animals.

These valves appeared as a better solution to mechanical valves, and due to enhanced physiological hemodynamics and reduced platelet adhesion, these valves do not induce thromboembolic complications, and thus, do not require anticoagulant treatments.

However, even in this case there are some limitations related to the restricted durability (lack of mechanical strength), the appearance of structural deteriorations (associated with the morphological changes of valves during of decellularization and chemical fixation) and the extensive calcification. Valve calcification appear when the non-viable cells are incapable of repairing the deteriorated ECM and their fragments serve as nuclei for calcification; chemical fixation increases the flexural rigidity of bioprosthetic valves and locks them into one configuration, which does not allow the dynamic ECM arrangements necessary for normal valve function [72].

***Autograft valves*** are prepared using the patient’s own tissues, by replacing the aortic valve with the autologous pulmonary valve of the patient (Ross procedure) [73]. In the same procedure, for a rapid restoration of blood flow in the exit right ventricle tract, a cryopreserved homograft is implanted. It was demonstrated that the reconstruction success rate of the right ventricle outflow tract (RVOT) is about 90% for children at 12 years [74]. The reconstruction of RVOT obstruction may involve resection of obstructing muscle bundles, creation of an RVOT patch, pulmonary valvotomy or valvectomy and pulmonary arterioplasty [75], and is often associated with mechanical and electrical abnormalities arising from non-contractile/non-conductive patch material [76,77]. The Ross procedure is accompanied by important advantages, such as the fact that the patients no longer need a life-long anticoagulant treatment, it reduces the risks of stroke or bleeding and it improves life expectancy, in comparison with the case of mechanical valves. These valves have also been shown to have a high regeneration/remodeling potential, have a similar physiological hemodynamic profile and, last but not least, be more cost effective than mechanical valves; such considerations make them a valuable choice for young patients [78].

***Allograft valves*** are used for pulmonary or aortic valve replacement, or for RVOT reconstruction, and are usually obtained post mortem. Although decellularized human pulmonary valves seeded with autologous endothelial cells [79] and human endothelial progenitor cells (EPCs) [80] demonstrated very good hemodynamic performance and functionality, in the case of younger patients, the risk of structural deterioration of the valve (owing to cryopreservation and thawing process) ranges from 71 to 87%, at 10 years [81]. Moreover, in comparison with mechanical valves, the allografts demonstrated an improved hemodynamic profile, low thromboembolic risk and low immunogenicity [82]. It was also shown that the decellularized allografts are a much better alternative to conventional cryopreserved valve grafts, due to a lower degeneration rate (>10% and 30%, respectively, after 5 years) and since cryopreservation procedures lead to surface and structural damages [83,84,85]. There are also major disadvantages related to this type of valves, such as their low availability considering the limited number of human organ donors, low durability because of residual immunogenicity and high possibility of their calcifications, which implicitly causes degeneration of the valve structure [86,87,88]. Considering all these aspects and the fact that this type of valves is usable for a limited period of time (only 30-40% are still functional after 20 years from implantation), it can be concluded that they are not recommended for elderly patients [13,89].

***Xenograft valves*** are biological grafts derived from animals, usually porcine and bovine, with a high usage due to a limitless supply. The porcine valve has the advantage of adequate anatomic structure and unlimited availability, while bovine pericardium has elastic properties in accordance with anatomical geometries, due to a higher amount of layered structural proteins [90,91,92]. Several xenografts have been developed in the idea to solve the problem of valve calcification and avoid the risk of postoperative mortality. As a result, four main types of valves have been proposed, which are classified into: stented, stentless, sutureless and transcatheter (percutaneous) valves (Figure 4).

A promising alternative tissue source for producing artificial leaflets is the bovine pericardium, which was treated with glutaraldehyde and then was mounted on Delrin flexible stent (Ionescu-Shiley valve), in order to achieve a synchronous opening of the three leaflets in the *stented valve*. However, after only 5 years of usage, structural valve deterioration was observed, caused by the rupture of the leaflets because of their movement within the stent, which led to severe aortic regurgitation [67]. Several models were designed (St. Jude Trifecta, Sorin Mitroflow, Carpentier-Edwards Perimount Magna), either by modification of the suturing technique of the pericardium onto the stent or by introducing different types of stents, more flexible or thinner, but without important improvements in the mortality risks. The treatment proposed in order to increase the stability over time of this type of valve, consisted of the following steps: (i) their washing to remove soluble proteins by using different surfactants (sodium dodecyl sulfate (SDS), TritonX-100, Tween 20, etc.), (ii) sodium periodate denaturation of structural glycoproteins and mucopolysaccharides, (iii) neutralization with ethylene glycol and (iv) one of the most effective methods of reducing tissues antigenicity, glutaraldehyde (GA) treatment for crosslinking the remaining free amino groups of amino acids [93,94]. Usually, xenografts are decellularized in order to avoid the activation of the recipient’s immune response, so the animal cells are removed from the graft, but the ECM is preserved in order to provide to the remaining scaffold the original anatomical structure and the adequate support for the seeding with the cells of the patient, after implantation [77,95]. The elimination of valve interstitial cells (VICs) determines the degeneration of prostheses, both in vitro and after implantation. For improving the mechanical properties of the xenograft, its cross-linking reaction with GA is a process where proteins are cross-linked and collagen fibers stabilized, conferring the graft with tensile strength, elasticity and resistance to degeneration [96]. However, the cross-linked grafts do not allow matrix-metalloproteinases (MMPs) degradation, which interferes with the remodeling process [97]. In addition, fixation with GA determines structural valve deterioration, towing to the host’s immune response and its further calcification. An optimal modeling process could be found by balancing the matrix formation process and graft degradation process.

Another option in order to minimize the structural valve degeneration has been the development of *stentless valves* (Medtronic Frestyle, Edwards Prima Plus, St. Jude Medical Toronto SPV), both porcine and pericardial, made without any stent or sewing cuff, and which are smaller in size than the stented valve. Two techniques are frequently used to implant these valves: the coronary technique and the complete root. While the first technique has an increased risk of valvular insufficiency because of changes in valve shape, especially for patients with calcified aorta, the second technique allows complete replacement of the root, thus restoring the physiological function and determining excellent hemodynamics, beneficial to patients [98]. Although they have improved hemodynamics, their use is difficult considering that they require a longer implantation time and specific surgical skills, and in addition, they have a high rate of perioperative aortic regurgitation because of a discrepancy between the valve annulus and the native sino-tubular junction [67]. The *sutureless valve* (Livanova Perceval S, Edwards Intuity and Enable 3F) solved important problems, such as the reduction of surgical traumas and wound complications, as well as a reduction of the cross-clamp and cardiopulmonary bypass time, especially for elderly patients. With regards to children and young adults, it was demonstrated that the Ross procedure has superior performances compared to the mechanical, homograft and bioprosthetic valves [99].

The most advanced and rapid development in cardiac surgery was recorded by the *transcatheter valves,* with technologies that include the design progress of valves, stents and catheter in order to give minimal surgical interventions (catheter-based devices) through an endovascular approach, without requirement of a cardiopulmonary bypass or cardioplegia [100,101,102]. There are different types of transcatheter aortic prosthesis, exemplified below [67]:The balloon-expandable valve (Edwards SAPIEN) was further subjected to an anticalcification process, consisting of GA fixation and a phospholipid extraction, and an additional “mildheat” treatment, which removes the unstable GA molecules;The self-expandable valve (Medtronic Corevalve) was initially made of bovine pericardium, but was later developed with a lower profile device, by using a more flared outflow design and a porcine pericardium.

These valves present a unique expandable frame, anti-calcification properties, improved hemodynamic performance and durability, thus reducing the likelihood of reoperation.

However, several limiting factors have been identified in bioprosthetic valves, such as paravalvular discharge, high possibility of vascular complications, risk of neurological events and complete atrioventricular block [67]. Moreover, it has been observed that, in a period of 20 years, the prosthetic aortic valves can present structural deterioration and immunogenicity, in a percentage of 55–70% of patients aged 60 years or even younger, which leads to the need to reoperate these valves [103]. Currently, available bioprosthetic valves have a restricted usage because they lack the capability of growth, repair, remodel and regeneration, even if they sometimes represent a superior alternative to surgery for a short term [104].

### 3.3. Polymeric Valves

Due to the increased risk of infection of mechanical valves and the risk of deterioration of bioprosthetic valves, new engineered scaffolds were designed, in different shape and compositions—*polymeric valves* [4,96]. The developing of scaffolds with the optimal characteristics, such as strength, rate of degradation, porosity and microstructure, as well as their shapes and sizes, is more readily and reproducibly controlled in polymeric scaffolds [105]. Their unique properties, such as high surface-to-volume ratio, high porosity with very small pore size, biodegradation and mechanical properties, have drawn great attention. They offer distinct advantages of biocompatibility, versatility of chemistry and the biological properties, which are significant in the application of HVTE and organ substitution [20].

The polymeric heart valves can be made either from *synthetic polymers*, *natural polymers,* various combinations of these materials or by their combination with inorganic molecules, e.g., carbon-based materials, ceramics or metal-based materials [106]. The main types of polymeric materials used in HVTE are presented schematically in Figure 5.

Since the 1950s, different synthetic polymers, such as polyethylene (PE), PGA, PLA, PLGA, PU, PCL, poly(methyl methacrylate) (PMMA), poly(vinyl alcohol) (PVA), polyhydroxyalkanoates (PHA), polyhydroxyoctanoate (PHO), poly(glycerol sebacate) (PGS), polytetrafluoroethylene (PTFE), etc., have been systematically investigated and the results were included in comprehensive literature studies [18,33,61,103,107,108,109].

The first choice regarding the use of polymers in the development of polymeric heart valves was polyethylene (PE), followed by poly(methyl methacrylate) (PMMA), both being selected for their robust mechanical properties [61]. Over time, it has been demonstrated that the advances in synthesis methods and structural modification have developed synthetic polymeric scaffolds with controllable structure, reproducible properties, together with biocompatibility, biostability and anti-thrombogenicity [64]. However, there are still limitations of these scaffolds related to cell adhesion and tissue reorganization.

Alternatives to synthetic polymers are the *natural polymers,* a preferred choice due to their structure, which is more similar to the components of ECM, with specific binding sites for cells, with a tailored biodegradation rate and an excellent biocompatibility [106,110]. Even if the scaffolds from natural polymers demonstrated their tissue remodeling capacity and structural in vivo durability, they have some drawbacks related to their relatively weak structure and poor mechanical properties. In order to increase the mechanical strength required to withstand the hemodynamics of stress in the cardiac environment, natural polymers have often been used together with the synthetic ones, to obtain the so-called *composite scaffolds*.

Some of the advantages and disadvantages of the polymeric scaffolds used in HVTE are presented in Figure 6.

A detailed discussion of the advances registered in the use of natural polymers in heart valve tissue engineering, taking into account the principles of scaffolding, the cells type and source as well as the variety of strategies, will be presented later in this review.

## 4. Heart Valve Tissue Engineering: Cells and Strategies

The highly complex architecture of heart valves includes an extracellular matrix (ECM) populated by *valvular interstitial cells* (VICs) and encapsulated by *valve endothelial cells* (VECs). All these are in a continuous reorganization, as a response to the changes during the cardiac cycle.

ECM is an extremely organized network, composed of three closely linked layers, arranged according to the blood flow, with unique properties that vary continuously throughout the cross section of the leaflet (Figure 7) [111], namely:-The *fibrosa layer* is located near the outflow surface and is made of collagen (COL) and represents densely aligned fibers that ensure the primary strength of the valves;-The *ventricularis layer* is located on the opposite surface of the entrance and is made of elastin (EL), with an important role in stretching and retraction during the cardiac cycle;-The *spongiosa layer* is located between the two layers mentioned above and is made of proteoglycans (PG)—glycosaminoglycans (GAG), with the role of loose connective tissue to facilitate the relative movements of the adjacent layers.

The quantity, quality and the structure of ECM depend on the viability and function of the VICs, this cell–matrix interaction being determined by a dynamic and complex mechanical stress state during every cardiac cycle [113]. Cell adhesion on the surface of ECM is mediated by the ECM components of the valve leaflet and consist of small amino acid sequences that mediate cell attachment, the most popular being the arginine-glycine-aspartic acid (RGD) domain [114].

While the ECM plays a critical role in the structure–function relationship of the valve, the VICs cells have an important role in preserving its architecture for functional biomechanics and maintaining homeostasis and also have a crucial role in some pathological valve processes. Moreover, the valve cusp is encapsulated by a single cell layer of VECs (Figure 7A), which creates a functional barrier between the blood and the inner tissue of the valve, acting as protection against physical and mechanical stress of the hemodynamic environment, and continuously communicating with VICs for regulating their phenotype [111].

The most numerous valvular cell types are VICs, which present particular characteristics and functions depending on the environmental conditions, and can be classified as: embryonic endothelial progenitor cells (eEPCs), quiescent VICs (qVICs), activated VICs (aVICs), progenitor VICs (pVICs) and osteoblastic VICs (obVICs). At different cycles of development, VICs show different phenotypes. In adult heart valves cultured in situ, VICs are quiescent and display a fibroblast-like phenotype, characterized by the presence of vimentin, and very low levels of α-smooth muscle actin (α-SMA), metalloproteinases (MMP-13) and SMemb (non-muscle myosin heavy chain) [113]. In contrast, in heart valves cultured in vitro, 50–80% of VICs isolated express high levels of myofibroblastic markers such as αSMA [115].

Biomechanical and biochemical factors have an important role in VICs response, so that VICs from aortic and mitral valves are more rigid than those from pulmonary and tricuspid valves, which suggests that VICs respond to local tissue stress by altering their stiffness [105]. VICs have a fusiform, ellipsoidal shape and contain a large amount of cytoplasm, rich in mitochondria, rough endoplasmic reticulum and exocytic vesicles. VECs have the role of maintaining a nonthrombogenic blood–tissue interface, for the transport of nutrients, regulating immune and inflammatory reactions and ensuring the transduction of biochemical and mechanical signals in the heart valve. Additionally, they have a cobblestone-like morphology and are aligned perpendicular to the blood flow direction and parallel to the collagen fibers from ECM [116].

Engineering of heart valves is greatly influenced by the type of the used cells, and the three most frequently used cell types TEHV are: *xenogeneic* (from a different species), *allogeneic* (same species) and *autologous* (same living being). If in the case of autologous cells, they have a high activity and are best suited for use in TEHV, allogeneic and xenogeneic cells invoke an immune response and cannot be used without immunosuppressive therapy [117].

However, it has been observed that analogous cell types from animal and human sources showed almost identical phenotypes in TEHV, which led to the possibility of using cells from animal sources in both in vitro and preclinical research. Cells from animal sources have several advantages, such as wide availability, being cheaper and not being subject to the same level of safety and ethical regulations as human cells. Another advantage of animal cells over human cells is the fact that they can be isolated from all four valves of the heart, while in humans, it is usually obtained from a single valve [118]. The cells used in TEHV, both from animal and human sources, are: mesenchymal stem cells, valvular interstitial cells, valvular endothelial cells, endothelial cells, miscellaneous cells and fibroblasts [117].

Starting from various *scaffolds* (1), including autografts, allografts, xenografts and polymeric scaffolds, and from different types of *cells* (2), appropriate to maintain and remodel the ECM, several *strategies* (3) have been established, in order to obtain living tissue valve replacements that can function like the native heart valve.

These TEHV strategies can be classified into: in vitro *TEHV*, in vivo *TEHV* and in situ *TEHV* (Figure 8).

Regarding the scaffolds used in TEHV, it should be mentioned the fact that allographs and xenografts are pre-processed by decellularization, to ensure immunocompatibility and a standard availability of valve tissues, mostly preserving the integrity and functionality of the ECM. The use of bioresorbable polymers in TEHV has attracted special attention due to the possibility of quickly manufacturing scaffolds with reproducible architectures, with controllable degradation rates and mechanical and chemical properties adapted to the desired purpose [78,82,119]. In addition, these scaffolds have the advantage of being absorbed and metabolized by the body.

In vitro *TEHV* strategy consists of the incorporation of autologous cells into a bioresorbable scaffold, which can be either biological or polymeric. This cell–scaffold system is usually cultured in a bioreactor to allow ECM deposition and to promote the new tissue synthesis with an adequate elasticity and strength for implantation [120]. To in vitro cellularize the scaffold before implantation, autologous cells are used with a view to preventing an immunogenic response, and these are [103,121,122,123,124,125]:(Myo)fibroblasts isolated from harvested vascular or dermal tissues;Mesenchymal stromal cells (MSCs) from bone marrow or adipose tissue;Prenatal or early postnatal sources of MSCs, where cells are harvested before or immediately after birth and used toward the synthesis of autologous valve tissue for replacement in early childhood;Amniotic membrane sources of MSCs (AM-MSCs);Amniotic fluid sources of MSCs (AF-MSCs);Chorionic villi sources of MSCs (CV-MSCs);Umbilical cord sources of MSCs (UC-MSCs), from the cord blood, Wharton’s jelly or perivascular tissue;Stem cell (iPSC)-derived endocardial cells with the potential to provide VIC-like cells by undergoing endothelial-to-mesenchymal transition, with the best potential to obtain the native VICs population, compared to other mesenchymal cells.

The optimal biological scaffolds, from a geometrical and hemodynamical point of view, are decellularized heart valves (allogenic or xenogenic). However, they also have an important number of negative effects, such as microstructural changes and altered protein composition, as a response to the cryopreservation process or decellularization, the limited availability of human tissue, the residual immunogenicity of animal tissue and the limited cellular infiltration [23]. Both synthetic and natural polymer-based scaffolds offered an attractive solution for TEHVs achievement, due to their unlimited availability, their tunable architectures and mechanical properties, and their inherent lack of xenogeneic disease transmission [126]. In addition, so far, in vitro TEHV has not advanced into routine clinical use.

The in vivo *TEHV strategy* uses the body’s ability to encapsulate foreign material and use fibroblasts to produce ECM proteins. In this sense, a valve-shaped mold is implanted subcutaneously, and this is covered over time by a fibrous tissue, which is then removed and used as a replacement valve. Even if the method seems accessible, unfortunately, apart from obtaining an adequate geometry of the valves, there is no control over the cells present in the tissue, nor over the ECM composition or its mechanical properties [127].

The in situ *TEHV strategy* consists of the direct implantation of an acellular resorbable scaffold, which can be either biological or polymeric, and which is designed to induce the potential for endogenous regeneration, directly at the functional site of the valve. In this type of strategy, the scaffold must ensure an optimal environment for the adhesion, differentiation and growth of the host cells, to support the formation of the new tissue, while the controlled degradation of the initial scaffold takes place, and that the newly formed tissue has mechanical properties similar to those of native functional tissue [128]. Moreover, the scaffolds that are used for in situ TEHV may be either newly fabricated (natural or synthetic polymers) or decellularized from native valves or bioreactor grown valves. For synthetic polymers, an alternative method to enable them to mimic native heart valves are their biofunctionalization by the incorporation of peptides, proteins or recognition sequences [127]. In the case when TEHV is grown in vitro and then is decellularized before further implantation, the choice for an autologous cells source is not absolutely necessary, because the tissue produced by allogeneic cells is immunocompatible, if adequately decellularized. Other cell sources may be the UC-MSCs cells and the induced pluripotent stem cells (iPSCs), which offer different advantages, in terms of accessibility, expandability and capacity for tissue synthesis [103]. To date, the in situ TEHV strategy recorded the highest progress, on the basis of different in vivo studies in animals and with delivery of the valve using transcatheter implantation, showing encouraging results [46,129,130].

The most relevant experimental approaches related to in vitro and in situ TEHV, together with cell source, the type of scaffold and the main results of the studies, are presented in Table 3 for the aortic valve replacements and Table 4 for the pulmonary valve replacements. 

## 5. Natural Polymer-Based Scaffolds for Heart Valve Tissue Engineering

Heart valve tissue engineering scaffolds based on natural polymers have the advantage to be prepared from environmentally friendly, renewable and low-cost raw materials, with appealing properties for biomedical applications, such as biocompatibility, biodegradability and intrinsic cellular interaction [20,154,155,156,157]. Although natural polymers provide excellent cell attachment and growth, they have many disadvantages, such as immune response problems or poor mechanical properties [16]. All these will be discussed in detail for each category of natural materials (i.e., polysaccharides and proteins), taking into account different examples for each polymer.

### 5.1. Polysaccharide-Based Scaffolds for Heart Valve Tissue Engineering

Polysaccharides are the most abundant biomaterials in nature that meet several criteria for eligible supports for tissue engineering, which include biocompatibility, biodegradation and the ability to support cell development [158,159]. Due to their biological properties and their structural and functional similarities to ECM, it is reasonable to use them in tissue engineering [160,161,162]. In combination with appropriate cells or bioactive molecules, the polysaccharides become an important asset to promote heart valve tissue regeneration [163]. Their applications for heart valve tissue engineering are vast and varied, and approximately 70% of all studies in this field focus on chitosan, alginate, hyaluronic acid and cellulose, respectively [160]. Table 5 presents several examples from the multitude of applications in valve engineering, for each of these polysaccharides.

#### 5.1.1. Chitosan-Based Scaffolds

Chitosan (CH), a naturally occurring linear polysaccharide obtained from chitin by alkaline deacetylation, is an attractive material for tissue engineering due to its unique properties, such as biodegradability, biocompatibility, non-toxicity, anti-bacterial effect, hydrophilicity and structural similarities to glycosaminoglycans (GAG), which is a major component of ECM [154,184]. The use of chitosan in HVTE has not been as extensively studied compared to other synthetic/natural materials [185], however there are numerous results that attest its success in this field, which are briefly summarized in Table 5 and described in detail below.

Specific adhesive properties of unmodified chitosan films and their potential as heart valve substrates against the native valve endothelial cells (VECs) were evaluated by comparison with other three biodegradable polymers, i.e., gelatin (Gel), poly(lactic-co-glycolic acid) (PLGA) and a polyester from polyhydroxyalkanoates family (PHA) [164]. Tissue culture polystyrene (TCPS) and Gel were used as positive controls, their cell behavior being extensively known [186,187]. The analysis of the cell growth (AlamarBlue Assay) revealed, after 7 days of incubation, a preferential order, namely TCPS > (Gel, PLGA, CH) > PHA. The VECs morphology variation on different substrates was evidenced by fluorescence imaging (Calcein AM), as seen in Figure 9.

On both positive controls, TCPS and Gel, the VECs showed cobblestone morphology with the formation of a typical confluent monolayer of endothelial cells (ECs). In contrast, on PLGA and CH substrates, VECs tended to be less spread and more elongated, with a more typical morphology for fibroblast/SMCs, and the cell–cell junction is missing. In addition, VECs on PLGA were slightly more spread with a uniform distribution, while on CH, they were often grouped in star clusters. The cells on PHA showed the lowest amount of spread of all substrates, with many cells remaining spherical and scattered, even after 6 days of culture [164].

As a consequence of the low performance of unmodified CH, an attempt to precoat its surface with adhesive proteins (fibronectin and mouse laminin) was performed, this being one of the simplest methods to enhance VECs adhesion and growth. However, the presence of protein coatings on CH leads only to modest improvement of VECs growth, the cells displaying low amounts of spreading and altered elongated morphology. It was concluded that poor protein adsorption to CH is, in fact, the cause of reduced cell growth on its scaffolds. In the next step, composite films based on CH and collagen were proposed as an alternative method to those tried before. Indeed, the composite films with collagen type IV support enhanced the growth of VECs compared with CH alone and VECs morphology was superior to the case of chitosan, with or without adhesive protein precoating. These were some preliminary attempts, but they still showed that chitosan combined with the appropriate protein can be a promising substrate for valve tissue engineering [164]. Another method used to increase the cellular adhesiveness of chitosan was loading chitosan nanoparticles with basic fibroblast growth factor (bFGF) and poly-4-hydroxybutyrate (P4HB). This complex was further used as a coating for decellularized porcine aortic heart valve leaflets to obtain a biomatrix-polymer hybrid scaffold [165]. Hybrid scaffolds were seeded with mesenchymal stem cells (MSCs) isolated from bone marrow of tibia and femur of adult S-D (Sprague–Dawley) rats. Two types of hybrid scaffolds were created: the “hybrid scaffolds bFGF” using solutions of P4HB and bFGF-loaded chitosan nanoparticles and “hybrid scaffolds” using only P4HB and chitosan nanoparticles, without bFGF. The morphology and ultrastructure analysis showed good cell–scaffold adhesion and growth of MSCs, leading to complete repopulation of both the valve leaflets and hybrid valve leaflets with bFGF. Of course, these were attributed to the stimulatory effect of bFGF on the proliferation of MSCs. The bFGF/chitosan/P4HB fibers randomly deposited on the decellularized valve leaflets surface and formed a membrane structure with uniform thickness, which firmly combined with the surface of the decellularized valve leaflets. This is the effect of the good biocompatibility of P4HB and chitosan, in particular of the chitosan, which brings some improvements related to the hydrophilicity and the presence of functional groups that are beneficial to the cell–P4HB–decellularized valve leaflets interaction. Thus, the hybrid valve leaflets (bFGF), fabricated for the first time using the electrospinning technique, could be useful for the generation of viable, functional heart valve prostheses [165].

Albanna et al. [188] chose to covalently immobilize heparin onto extruded chitosan fibers using 1-ethyl-3-(3-dimethylaminopropyl)-carbodiimide (EDC), in order to improve the cell adhesiveness of chitosan. The heparin was chosen due to its known ability to modulate the growth factors and binding proteins. All factors involved in the fiber extrusion technique, such as the temperature or the concentrations of acetic acid and ammonia solution, along with the degree of heparin crosslinking, can influence almost equally the fiber diameters, strength and stiffness. Chitosan-heparin fibers promoted the attachment and growth of valvular interstitial cells (VICs) from the first day of culture and continued until day 10, achieving a cell viability of 95% [168].

Chitosan coatings were applied on bovine pericardium (BP) tissue in order to improve its biocompatibility and alleviate calcification, but also to confer antimicrobial activity to collagen tissue [166]. The BP samples were immersed for 2 h into a chitosan–carbonic acid solution, in order to adhere spontaneously to collagen tissues. The control samples, named “dummy” samples, were also prepared under the same conditions, treating collagen tissues in pure carbonic acid without chitosan (Figure 10).

Two samples of BP tissues were implanted subcutaneously into Wistar rats; each rat received simultaneously a control sample of BP stabilized by GA, and the second sample consisting of either a “dummy” sample or a CH-coated sample. No traces of any inflammatory reaction were observable after 4 months since the implantation. Test cells, using mouse fibroblast line NIH/3T3, showed that the positive control induced “severe” reaction (more than 70% lysis), whereas reaction was absent for both the negative and the reagent controls. Some structural changes, consisting of the collapse of macropores, occurred in the BP matrix during deposition of chitosan from solutions, most probably related to the influence of high pressure of the solvent. CH effectively “glues” the walls of the collapsed pores and the BP matrix becomes more solid, having fewer pores and voids. This results in improved stress–strain properties, the BP tissue being more flexible and less rigid, and therefore, the risk factor of a fatigue failure of the bioprosthesis is reduced. Moreover, the coating of BP with CH dramatically mitigates the calcification process and the adhesion of some strains of both Gram-positive and Gram-negative bacteria is suppressed [166].

A chitosan (CH) complex with gelatin (Gel) and polyurethane (PU) was used to obtain a thin and robust trilayered structure heart valve leaflets [167]. PU is widely used in cardiovascular applications due to its elastic property and good compatibility with blood components [189], while Gel is biodegradable, nonimmunogenic and promotes cell adhesion and proliferation [190]. The trilayered nanofibrous structure consisted of PU nanofibers as the middle layer and the Gel-CH complex as the outer layers, on both sides of PU fibers. The biocompatibility and cell retention ability of the Gel-CH-PU substrate was compared to collagen coated-pericardium and PGA-PLA copolymer, a biocompatible material used in cardiac tissue engineering applications. The ECs chosen for this study were isolated from ovine carotid arteries (OCAs). They adhered onto all three materials one day post-seeding, with the cells being flattened and spreading across the surface and adopting typical cobblestone morphology. However, a significant increase in cell coverage was observed as early as one day post-seeding for the Gel-CH-PU complex, to which the cells preferentially adhered, and this behavior continued also in the next 14 days. Moreover, when ECs seeded on the Gel-CH-PU were subjected to shear stresses, mimicking the pulsatile flow to which valve leaflets are continuously exposed during their operation, the cells were able to withstand to shear stresses ranging from 0.062 to 0.185 N/m^2^, for up to 3 h. Only the maximum shear stress causes little changes in the cell coverage area, i.e., 88.10 ± 7.11% after 1 h and 78.83 ± 12.49% after 3 h, respectively. In contrast, PGA-PLA and collagen-coated pericardium suffered from significant cell loss with increasing shear-stress and exposure time (i.e., 35% decreases after 3 h exposure time for the PGA-PLA group). The results highlighted once again the feasibility and attractiveness of the electrospinning technique to be used in the fabrication of thin and robust layered scaffolds and also to create complex structures, such as the trilayered heart valve leaflets [167].

Electrospun chitosan fibers were involved, together with polycaprolactone (PCL), in the fabrication of biohybrid nanofibrous scaffolds by coating a decellularized bovine pericardium (DBP) [169]. PCL provides an appropriate mechanical strength required for valve tissue engineering and CH improves the adhesion to DPB tissue. CH-PCL nanofibers were obtained in a customized electrospinning device at room temperature (27–32 °C) and a voltage of 15 kV. Aligned (A) and random (R) nanofibrous DBP-PCL-CH biohybrid scaffolds, with favorable structural and biomechanical properties for TEHV, were developed. Previous positive results have been considered regarding the superior uniaxial mechanical properties and slow degradation ability of aligned nanofibrous polymeric scaffolds compared to non-aligned (random) fibrous scaffolds [191,192]. Human valve interstitial cells (hVICs), isolated from patients undergoing valve replacement surgery, were seeded on these biohybrid scaffolds. SEM images and fluorescence micrographs correlated with live cell imaging using DiI labeled cells, presented in Figure 11, reveal better cellular attachment onto the biohybrid scaffolds than DBP, and also high alignment of cells along the polymeric fibers (Bio-hybrid A) compared to the randomly electrospun samples (Bio-hybrid B). The cell viability was for all the scaffolds around 90%, but on the biohybrid scaffolds, the cells were able to metabolize the tetrazolium dye, indicating more viable hVICs. In addition, aligned biohybrid scaffolds demonstrated significantly improved uniaxial mechanical properties with optimum pore and fiber diameter that finally slowed down the degradation rate [169].

Even though the chitosan has numerous attractive properties for the HVTE field, the unmodified chitosan also has some disadvantages, namely its poor cell adhesiveness and weak mechanical properties. These drawbacks can be overcome by different approaches. For instance, the cell adhesiveness can be improved by using appropriate proteins, loading with basic fibroblast growth factor or by covalent immobilization of heparin, which has the ability to modulate the growth factors and bind proteins. On the other hand, the mechanical properties can be improved by incorporating chitosan into various hybrid composites along with materials with good mechanical properties (i.e., PU, PCL). All these measures have proven to be very suitable in making chitosan a promising material for obtaining substrates for valve tissue engineering.

#### 5.1.2. Hyaluronic Acid-Based Scaffolds

Hyaluronic acid (HA) has the major advantage of being an omnipresent component of ECM, the spongiosa layer of the valve leaflets containing a considerable amount of HA. Other advantages of this large linear polysaccharide that are worth mentioning are its ability to recognize cells through cell-surface receptors and, the most important, its functional groups enable the biological and mechanical properties of the scaffold to be modified. Although HA is an ideal scaffold for cell encapsulation, it does not exhibit the mechanical properties necessary for the physiological function as a scaffold for heart valve leaflets, and thus, does not meet the design criteria for TEHV. However, hybrid scaffolds or chemically modified scaffolds can mitigate some of these issues and can be a step further in developing a TEHV [185].

The biocompatibility of HA is one of the key factors that guarantee its efficiency as a support for cell growth. Masters et al. propose the biocompatibility evaluation of photopolymerizable HA-based materials as scaffolds for VICs, the most prevalent cell type in native heart valves [170]. VICs were encapsulated into methacrylate modified-HA (HA-Me) hydrogels in order to investigate the possibility of the products enzymatic degradation to affect VICs’ behavior. Moreover, HA-Me was copolymerized with poly(ethylene glycol) diacrylate (PEG-DA) to expand the properties of these hydrogels. Photopolymerization was chosen as a cross-linking method as it occurs under relatively mild conditions and enables the encapsulation of cells within the scaffolds. Swelling properties, degradation time and mechanical stiffness increase upon copolymerization of HA with PEG-DA. When VICs were exposed to HA degradation products, there was a significant increase in cell proliferation and elastin production. These two processes were highly dependent on the HA molecular weight, VICs proliferation recording the most prominent increase with lower molecular weight LMW HA (˂27,000 Da). Furthermore, after 3 days of culture, LMW HA significantly stimulated the production of elastin by VICs, while after 20 days, there was a considerable production of ECM. These results have shown that HA-based materials are biologically active, and their potential for use in 3D valve tissue generation is promising [170].

Later, Camci-Unal et al. [171] evaluated the surface cell adhesiveness of these methacrylated HA-based materials, proposing an innovative strategy involving the covalent conjugation of CD34 antibodies on HA hydrogels’ surface to selectively capture the endothelial progenitor cells (EPCs) and promote endothelialization of tissue constructs. Two different CD34 antibody concentrations (10 and 25 μg/mL) were used to obtain the antibody-modified hydrogels and the highest number of EPCs was obtained for 25 μg/mL CD34 at 1 h (52.2 ± 5.0 EPCs/mm^2^). However, HA-CD34 hydrogels did not promote spreading of EPCs, maintaining their round shapes even 48 h after seeding. Thus, the next step was to add 2% (w/v) of gelatin methacrylate (Me-Gel) to HA hydrogels. Indeed, the fluorescent images for Calcein AM and Phalloidin staining images identified adhered and elongated EPCs on CD34-HA-Me-Gel, showing a significantly higher spreading for cells [171].

These promising results have allowed the transition to more complex hybrid scaffolds based on HA and Gel, with better biological and mechanical control over its scaffolds. Duan et al. [175] developed more complex hybrid hydrogels based on photocrosslinkable modified-HA materials using two different strategies for VICs encapsulation. There were six types of hydrogels, namely methacrylated HA (Me-HA) and methacrylated-oxidized-HA of two different molecular weight (MeO_0.5_HA and MeO_0.1_HA), with and without addition of Me-Gel, respectively. VICs were encapsulated as individual cells (homogeneous encapsulation) to study VICs spreading and phenotype and as a single cell cluster (spheroid encapsulation) to study cell migration into hydrogels. Encapsulated VICs were alive (green) after 14 days of culture, as seen in Figure 12. The hydrogels without Me-Gel are characterized by high mechanical stiffness, so they cause a delay in the spreading process of VICs, which showed significantly higher cell circularity. The presence of Me-Gel stimulates the cell spread, proliferation and migration from encapsulated spheroids. After 14 days, the cell circularity in hydrogels with low stiffness (with Me-Gel) was significantly lower than that in stiffer ones (without Me-Gel). Moreover, extensive VICs spreading was found only in hydrogels with Me-Gel, especially in Me-HA/Me-Gel, where cells presented spindle-like morphology.

Cell viability during 14 days of culture, measured by using the MTT assay, was over 75% for all hydrogels. Significant differences in terms of cells proliferation rate are observable on day 14, where the MTT absorption for softer hydrogels was significantly higher than for the stiffer ones, confirming the significant role of Me-Gel in increasing cell proliferation. These findings are very important when it comes to the rational design of the hydrogels to control the morphology, phenotype and function of encapsulated VICs [175].

One year later, Eslami et al. [176] integrated electrospun microfibers of poly(glycerol sebacate)/polycaprolactone (PGS-PCL) within Me-HA/Me-Gel hybrid hydrogels. The aim was to create a composite biomaterial that combines the advantageous ECM-mimicking properties of hydrogels with the mechanical properties of PGS-PCL elastomeric microfibers in order to obtain similar cellular environment and mechanical properties of native heart valve tissue. The electrospun fibers were integrated into a hydrogel by simple immersion in its precursor solution and then cross-linked by exposure to UV light. Sheep mitral valve interstitial cells (MVICs) were used to test the suitability of the composites and various properties, such as swelling ratio, stiffness, porosity, enzymatic degradation, as well as the in vitro analysis, have been determined. Initially, the MVICs assumed a rounded shape and began to spread over time, with a high level of cell viability (≥ 90%). The MVICs viability remains high (over 90%) even after the 21^st^ day and there is also a substantial increase in the number of cells. The SEM images of the scaffold in cross-sections reveal the cells at different depths, with non-homogenous distribution throughout the scaffold. On the PGS-PCL fibers alone, the cells are predominantly spread over their surface, possibly due to the dense fiber structure that temporarily limits cell infiltration. On the other hand, in composite structures, MVICs are present at different depths and are more attached to the fibrous component, due to migration of cells into the hydrogel component. All these advantages of microfibrous hybrid scaffolds, along with the preservation of mechanical properties (i.e., the Young’s modulus and ultimate tensile strength) make the composite hydrogel/PGS-PCL scaffolds a more suitable 3D structure for generating scaffolds for HVTE [176].

#### 5.1.3. Cellulose-Based Scaffolds

Cellulose possesses some attractive properties that make them suitable for tissue engineering, such as intrinsic biocompatibility, biodegradability, low toxicity or non-toxicity, low cost as well as the numerous intermolecular and intramolecular hydrogen bonds that give an excellent mechanical performance to its network [193,194,195,196,197,198,199,200,201,202]. However, when considering cellulose as a scaffold material, an important factor remains its biodegradation in vivo [203,204]. Cellulose is commonly known to be degradable in vitro by a certain category of microorganisms, but in vivo cellulose resorption does not occur due to the lack of cellulases in animals and humans [15]. Cellulose also has lower bioactivity compared to proteins, such as collagen, which show efficient cell growth and proliferation as a result of cell surface receptors [205]. In view of the above limitations, native cellulose is used relatively rarely in HVTE, although it is biocompatible and has good mechanical properties.

*Bacterial cellulose* (BC), a nanomaterial synthesized by certain species of bacteria, is a fascinating biopolymer with a unique combination of biological, physico-mechanical and structural properties that make it valuable for biomedical applications, including tissue engineering [206,207,208,209]. However, in some tissue engineering applications, the cells tend to grow but do not attach to the native BC substrates. Several strategies to improve the biocompatibility of the BC scaffolds have been proposed, some of them implying surface or porosity modification, and others the introduction of BC nanofibers into different composites [198,210,211]. For instance, the anisotropic nanocomposite made of BC and poly(vinyl alcohol) (PVA) was comprehensively studied by various researchers, since it possesses properties that fall within the mechanical range of heart valve tissues [179,180,181,182]. Millon’s research from 2006 [179] and 2008 [180], respectively, examined the possibility of using PVA–BC nanocomposites in cardiovascular tissue replacement, compared with porcine aortic heart valve leaflets. The nanocomposites obtained by the molding technique, with different concentrations of BC and PVA, were subjected to six freeze-thaw cycles, in order to evaluate the influence of the mixture composition and the number of cycles on the mechanical properties. The increase in modulus is evident when BC or PVA concentration increases, i.e., by adding 0.61% BC, it increased over 5 times (60% strain), and close to 4.5 times when the PVA concentration was increased from 7.5 to 15%. Additionally, an apparent increase in stiffness is observed with increasing the number of freeze-thaw cycles. Thus, for at least one type of PVA-BC nanocomposites, the stress-strain properties can be correlated with both the circumferential and the axial direction of the porcine aortic valve leaflets [179]. By applying an initial strain of 25, 50, 75 or 100% of the original length, there is a clear rising trend in stiffness as the initial strain is increased, and a better effect is observed in the longitudinal direction than in the perpendicular one. Depending on a particular application, a specific composition of the PVA-BC nanocomposite and processing parameters can be chosen to create a custom-designed biomaterial with the required mechanical properties of the tissue it is going to replace [180]. Similarly, Mohammadi and coworkers [181,182] used PVA-BC composites in the design of a trileaflet mechanical heart valve (MHV). The anisotropic PVA-BC composite, with 15% PVA and 0.5% BC, was processed for four thermal cycles and an initial strain of 75%. The aim was to mimic the non-linear mechanical properties and the anisotropic behavior of the porcine heart valves, applying a controlled strain, while undergoing low-temperature (−20 °C) thermal cycling. The composites have mechanical properties similar to those of the porcine heart valve, in both circumferential and axial directions. Even more, the ultimate tensile strength (UTS) > 50% and the higher stiffness in both directions make these composites a better match for the anisotropic behavior of heart valve (HV) than any isotropic biomaterials [182].

*Nanocelluloses* have attracted increasing attention in tissue engineering, especially as reinforcement fibers for polymeric hydrogel scaffolds, due to their nanoscale size and outstanding mechanical and biological properties. Additionally, the size and morphology of the nanocellulose fibers have been reported to have a positive influence on cell adhesion, proliferation and their chondrogenic differentiation [212,213,214]. *Cellulose nanofibrils* (CNF) from pineapple leaf were used to reinforce polyurethane (PU) films, in order to obtain hybrid composites with good long-term hemodynamic function, which did not fail because of biological degradation or fatigue, and maintained a low thrombogenic character on the surface [177]. The nanocomposites were prepared by compression molding, by stacking the CNF mats between PU films. The nanofibrils reinforced efficiently the PU film; thus, the mechanical properties of the composites were improved with increasing CNF content. The addition of 5 wt% CNF has proven to be optimal for obtaining more than satisfactory mechanical properties for prosthetic valves: the strength of the PU-CNF increased by 300% compared with neat PU, the modulus of elasticity (E) by more than 2600% and the stiffness by 2600%. No failure was reported after the accelerated fatigue tests onto prosthetic valves, for five out of five valves, representing the equivalent of 12 years cycles. The failure occurred only after the equivalent of approximately 13 years cycles for two of the five valves tested, while for three valves, it occurred after approximately 15 years, the equivalent of 608 million cycles [177]. Modification of *nanocrystalline cellulose* (mNCC) by TEMPO-oxidation, followed by covalent conjugation onto methacrylated gelatin (Me-Gel) backbone is a method that leads to hydrogels (mNG) with high biocompatibility, good mechanical properties and resistance under osteogenic media conditions [178]. Me-Gel is commonly used in tissue engineering having mechanical and excellent cell adhesion properties and also the ability to be used in bioprinted heart valves [131]. There is an increase of mechanical properties of mNG with increasing mNCC content from 0.2 to 2%, particularly in the strain energy, transition modulus and the elastic modulus. The incorporation of 2% mNCC resulted in increased strain energy, compared to Me-Gel, from 73.8 ± 32.7 kPa to 178.0 ± 65.8 kPa. Human adipose-derived mesenchymal stem cells (HADMSCs) seeded on the mNG scaffolds showed decreased α-SMA expression and increased vimentin and aggrecan expression, suggesting that the material displayed phenotypic properties found within the spongiosa of the heart valve. The incorporation of mNCC into Me-Gel enhanced the cells spreading in a concentration-dependent manner (Figure 13a,b), the percentage of cells spreading increased with increasing concentration of mNG [178].

Although the metabolic activity of the cells incorporated in mNG hydrogels was initially lower than the Me-Gel group, they still showed the greatest improvement between 7 and 14 days and reached the same level of Me-Gel alone (Figure 13c). Moreover, under osteogenic media conditions, mNG hydrogels showed lower expression of osteogenic genes, including Runx2 and osteocalcin, indicating resistance to calcification. Together, all these results identify mNG as an attractive biomaterial for HVTE applications [178].

The problem in the HVTE field arises especially with the long-term use of cellulose-based scaffolds, because of the tearing and calcification of the leaflets under high dynamic stress and the oxidative reactions in contact with blood components [215]. However, satisfactory results have been reported regarding the use of cellulose use in the manufacture of heart valves, after it has been subjected to some chemical modifications (see Table 5).

*Cellulose acetate* (CA) is a promising material in tissue engineering, thanks to the ability of CA-based scaffolds to support cells growth and enhance cell connectivity. Moreover, its biodegradability can be controlled by hydrolysis or enzymatic action with glucose as final product [12,216]. Unlike other biomaterials commonly used in HVTE (i.e., PLA-PGA, PVC, nylon, polystyrene, etc.), CA films are transparent and no autofluorescence, allowing the examination of cells by light and fluorescence microscopy [217]. Additionally, it could be easily molded, modified or blended with other polymers to improve the scaffolds’ integrity [218]. For instance, to create a biomimetic environment for cell development and to improve their adhesion and proliferation, an innovative method consisting of functionalization of the CA scaffolds surface was applied. Functional molecules, such as RGD peptides (Arg-Gly-Asp) and YIGSRG laminins (Tyrosine-Isoleucine-Glycine-Serine-Arginine-Glycine), immobilized through biotin-streptavidin bonds, have been used. The biofunctionalized scaffolds were further used as coatings for aortic metallic valves in order to create native valve anatomical structure [12]. The toxicity measurements by direct MTT assay at 1, 3 and 7 days, respectively, confirmed increased cell proliferation on functionalized scaffolds compared to surfaces without active molecules. After covering the surface of the valve leaflets with the CA-based nanoscaffolds, SEM images confirmed that cells grew successfully on the valve surface and the growth was strong even in the first 24 h. In addition, the bioactive factors not only increased the biocompatibility of the surface valve, but also provided controlled endothelization and therefore, a reduction in valve thrombosis. In this way, it was possible to combine the advantages of artificial valves with those of biological ones, obtaining more efficient valves that do not generate thrombosis after implantation, and thus, do not require the use of anticoagulants for life [12].

In summary, although the applications of cellulose-based materials in HVTE are quite few, compared to other known polysaccharides, there are certain properties that recommend these materials, especially their excellent mechanical strength, highly pure nanofibrils’ network structure and their inherent biocompatibility. If, in addition to all these, we add the ability of cellulose to be easily modified and also to offer special properties to composites in which it is incorporated, then we can affirm that cellulose materials deserve to be considered when it comes to obtaining scaffolds in HVTE.

#### 5.1.4. Alginate-Based Scaffolds

Alginate (Alg) has been widely used in tissue engineering as platform for encapsulation of a variety of cell types [219,220]. The common properties that recommend it in this field are its biocompatibility, low cost and easy implementation into 3D bioprinting process [221]. However, in the narrow field of heart valve engineering, Alg is not suitable by itself because of some limitations that appear especially in its long-term use, when there is evidence of alginate degradation in physiological conditions, determined by the exchange of divalent with monovalent ions [185]. Another major drawback of Alg appears from the large batch to batch variation in its biosynthesis, along with the high cost of its bacterial biosynthesis and the fact that it is not yet a scalable process. Furthermore, the hydrophilic properties of Alg make difficult the protein adsorption and cellular recognition. Consequently, chemical modification or combination with other materials that have good mechanical properties or slow degradation rate is required to obtain scaffolds with suitable properties for HVTE [222]. A proof in this sense is the suite of applications of alginate in valve engineering (see Table 5), some of them being presented in detail below.

Hockaday et al. [137] used a simultaneous 3D printing/photo-crosslinking technique for engineering a heterogeneous valve scaffold. The native anatomic aortic valve geometries were 3D printed using poly(ethylene glycol) diacrylate (PEG-DA) hydrogels supplemented with Alg. PEG-DA is a photo-crosslinkable elastomer with good mechanical properties, with slow in vivo hydrolytic degradation and can be functionalized with other precursors to encourage cell attachment [223,224]. Alg was incorporated into PEG-DA formulations to temporarily increase viscosity during the printing extrusion process. The precursor solution containing 10–15% *w*/*v* Alg and 0.2–2.0% *w*/*v* photoinitiator was injected into 4–8 mm diameter, 1 mm thick disc molds made with either 700 or 8000 MW PEG-DA. The solutions were then cross-linked by exposure to UV light. Valve interstitial cells from porcine aorta (PAVICs) were chosen as model cells to investigate scaffold biocompatibility. SEM images show that PEG-DA supplemented with 10% *w*/*v* alginate led to small pores (6.1–14.0 μm^2^) with sizes depending on the PEG-DA molecular weight. PAVICs-seeded scaffolds allow cells to grow along the entire surface of the conduits, but to a lesser degree on the root and leaflet interstitium. Additionally, high cell viability, about 91.3 ± 10.7% on day 1 and 100% on day 7 and 21, respectively, was observed, indicating that both the printing method and the resulting material were not cytotoxic [137].

Aortic valves with anatomical architecture using alginate/gelatin hydrogels (Alg/Gel) were manufactured by the 3D bioprinting technique [133]. The particularity of this study was the incorporation of cells in a regionally constrained manner, i.e., SMCs in the valve root and VICs in the aortic valve leaflets, respectively. VICs and SMCs were encapsulated into Alg/Gel hydrogels, and then each cell laden-gel was loaded into a syringe and directly extruded into disc molds. The printing accuracy was 84.3 ± 10.9%, the Alg/Gel hydrogels maintaining their geometry and mechanical integrity after extrusion and crosslinking. Cell viability after 7 days of culture, determined via Live/Dead staining, for both cell types, was over 80%, more exactly 83.2 ± 4.0% for VICs (Figure 14) and 81.4 ± 3.4% for SMCs, respectively (Figure 15). Moreover, the encapsulated SMCs expressed elevated α-SMA in a stiff matrix, while the VICs expressed elevated vimentin in a soft matrix. Thus, anatomically complex and heterogeneously encapsulated aortic valves can be manufactured using 3D bioprinting [133].

Alginate-based hydrogels can serve as coatings for bioprosthetic heart valves (BHVs) against calcification, the main cause for BHVs failure [183]. This study was the first research in this area that used Alg as a protective layer for BHVs, providing versatile and long-lasting anti-calcification properties. In addition, the biological neurotransmitter dopamine (Dop) was chosen to functionalize the Alg surface hydrogels, due to Dop’s strong binding property through a polydopamine composite layer, as a result of an oxidation-cross-linking reaction in water [225]. Bovine pericardium was treated by two different methods: (i) *control group*—crosslinking with glutaraldehyde (0.25% Glut/24 h + 1% Glut/24 h); (ii) *Glut–Dop–Alg group*—treatment with 5% Dop–Alg and then crosslinked with Glut. SEM images confirmed the formation of the multilayer coatings, with Dop-Alg coatings on both sides of the pericardium. Calcification dynamics were determined in vitro and in vivo, respectively. In vitro, a linear increase in calcification was observed, by day 3, both for the control and the Glut–Dop–Alg groups, when the calcium content was 2.313 ± 0.140 mg/L and 2.919 ± 0.252 mg/L Ca/mg tissue dry weight, respectively. By day 6, this value decreases dramatically in saline only for the Glut–Dop–Alg group to 0.725 ± 0.012 mg/L Ca/mg, verifying the initial hypothesis according to which a Dop–Alg coating serves as a protective layer for BHVs anti-calcification. Calcification dynamics in vivo were achieved by implantation of tissues of 1x1 cm dimensions in subdermal pockets in male SD (Sprague–Dawley) rats. The calcification data have been collected by day 20 and day 30 [183]. The calcium content constantly increased in the control group from 1.610 ± 0.124 mg/L on day 20 to 2.018 ± 0.135 mg/L on day 30. In contrast, for the Glut–Dop–Alg group, the calcium content first increased to 1.737 ± 0.124 mg/L on day 20 and then decreased to 0.675 ± 0.084 mg/L on day 30. Taking into account these in vivo results, it was demonstrated once again that the Dop–Alg coatings have a positive role in anti-calcification of BHVs.

As illustrated by the data presented above, by combining Alg with materials that have either have good mechanical properties and slow degradation (i.e., PEG-DA fibers) or can improve its protein adsorption and cellular recognition properties (i.e., methacrylated gelatin), alginate-based scaffolds with suitable properties (i.e., noncytotoxic, with elevated cell adhesion and mechanical integrity) for HVTE can be obtained.

### 5.2. Protein-Based Materials as Scaffolds in Heart Valve Tissue Engineering

Protein-based scaffolds have inherent biocompatibility, degrading under the specific action of enzymes and, thus, allowing cell-controlled tissue remodeling [96]. The most commonly employed proteins that may be suitable for HVTE are collagen and fibrin, but gelatin, elastin, keratin, silk fibrinoid or others may also be mentioned. Certainly, among all proteins, collagen is the most suitable in this regard, being the major component of ECM and providing most of the mechanical and tensile strength of the native valve. Moreover, all collagen types are biodegradable due to their protein nature and are poorly immunogenic mainly due to their homology across species [226]. Table 6 presents some results of relatively recent studies regarding the applications of protein-based scaffolds in HVTE, which will be further discussed in detail.

#### 5.2.1. Collagen-Based Scaffolds

The use of collagen (COL) as a biomaterial for TEHV involves a number of advantages, but some disadvantages as well. A major advantage of collagen is its peptide sequence Arg-Gly-Asp (RGD) recognized by cells, thus, resulting in the activation of several signaling pathways that promote proliferation, migration and prevent apoptosis. However, collagen also has several disadvantages, namely: (i) the lot-to-lot variation, from different animal sources or tissue samples; (ii) it can cause an immunogenic reaction, if originating from other species or even other humans, and antigens are present; (iii) it can activate blood coagulation pathway, through platelet adhesion and aggregation, when it comes in contact with blood [185]. Moreover, collagen does not meet the design criteria for HVTE by itself because of its poor mechanical properties, its temperature sensitivity and degradation during sterilization processes. Thus, its modifications either by crosslinking or a combination with other materials are required to be able to utilize collagen as a material for HVTE [238]. Some of these methods are presented in detail below.

The decellularization process of biological matrices is a common practice to obtain biological scaffolds based on COL, by selectively removing matrix components and creating more porous structures for cell repopulation. Through this preferred procedure, decellularized matrices retain their natural biological composition and the 3D architecture suitable for cell adhesion and proliferation [236,239]. Such a biological collagen-based scaffold was obtained by decellularization of porcine aorta tissue using sodium dodecyl sulfate (SDS) extraction, followed by EDC/NSH crosslinking route and treatment with elastase to remove elastic fibers [227] (see Table 6). After performing the stress-deformation analysis and in vitro enzyme digestion, the COL scaffolds turn out to have adequate mechanical properties and to be highly biodegradable. 3T3 mouse fibroblasts were seeded on COL scaffolds and cultured on a tissue rotator for 4 weeks, to evaluate cell adhesion, proliferation and infiltration properties. Live/dead assay showed that fibroblasts were able to adhere and proliferate on the COL scaffolds with excellent viability, suggesting that scaffolds were non cytotoxic and were inductive for cell growth. After initial attachment and proliferation on the surfaces, the fibroblasts were found to infiltrate the intramural layers of the scaffolds, the depth of cell infiltration increasing with culture time, so a 40 mm depth was obtained after 28 days of rotary cell culture. All these results indicated that COL scaffolds resulting after decellularization of vascular matrices have the potential to serve as scaffolds for tissue engineering [227].

Highly aligned, compacted collagenous constructs were manufactured by Shy and coworkers [228] in a specially designed bioreactor that involves application of both biomechanical (controlled static tension) and biochemical stimuli to increase cellular proliferation, matrix synthesis and mechanical properties of the constructs. The COL-based matrix, resulting from a mixture of solubilized COL and neonatal rat aortic smooth muscle cells (NRASMCs), were used to create mitral heart valves. The application of mechanical constraint (uniform tension during collagen compaction) determined the COL fibrils to align in the constraint direction, and also increase the cell content, stimulates their metabolism, and ultimately, led to stronger constructs. Although initially the constructs contained only collagen and cells, the presence of elastin inside the COL fibers, elastin sheath around the collagen core and proteoglycans at the interface of the COL fibers was also detected. These confirm that the cells entrapped in the constructs were metabolically active. Therefore, it is possible to improve the organization of collagen fibrils, which, together with elastin and newly synthesized proteoglycans, leads to improved structural integrity and stronger constructs [228].

Three-dimensional collagen matrix was manufactured using an innovative method based on rapid prototyping, a technique based on a 3D inkjet printer that allows the creation of a sacrificial mold in a series of layers [229]. A bovine Achilles tendon collagen type I solution was used to mold the collagen biological supports for VICs seeding. The behavior of VICs, regarding their ability to release proteolytic enzymes that are of significant importance in TEHV and also to synthesize their own matrix, was analyzed. The cells were seeded onto discs of 1%, 2% and 5% *w*/*v* collagen scaffolds, under static conditions, for 28 days. VICs remained viable during the 28 days of the experiment, as indicated by the MTS assay results. However, the cells proliferate more on the 1% w/v collagen scaffolds (193 ± 6%) than on the 2% or 5% *w*/*v* scaffolds (139 ± 7.6% and 132 ± 2.3%, respectively). The presence of the remodeling enzymes and matrix metalloproteinases (MMPs), as well the detection of ECM gene expression indicates that the VICs have the capacity to remodel the COL scaffold and to synthesize a new matrix.

Collagen mixtures with other proteins or polysaccharides have been the subject of many experimental studies in the last decade, in order to achieve complex heterogenous matrices and to evaluate the impact of their composition on the microstructure, biological and mechanical properties of the scaffolds. One of these studies involved a mixture of collagen (COL), elastin (EL) and chondroitin-4-sulfate (C4S), to obtain complex heterogenous scaffolds resembling each layer of the native heart valve, i.e., COL scaffolds, COL-EL scaffolds and COL-C4S scaffolds [230]. The scaffolds were prepared in polytetrafluoroethylene (PTFE) molds, by the freeze-drying technique, using suspensions with various compositions: 100% COL, 50% COL+50% COL, 20% COL+80% EL, 90% COL+10% C4S, 50% COL+50% C4S. As expected, the composition has a major impact on the microstructure and mechanical properties of the scaffolds, and also on cells proliferation (Table 7 and Figure 16).

All the scaffolds possessed approximately equiaxial pores, and the pore size increased with EL and decreased with C4S addition. An elastomeric behavior was exhibited under tension and compression and the mechanical properties (i.e., the tensile, compressive and bending moduli) decreased with decreasing COL content. However, considering the bending modulus value of the porcine aortic valve of 492 kPa [240], the mechanical properties of COL-scaffolds are still much lower than those of a native tissue, suggesting the need for better structural control and higher crosslinking density. The cytocompatibility was evaluated against the cardiosphere-derived cells (CDCs), which are able to differentiate into cells of the three cardiac lineages. CDCs showed good cell proliferation on the COL-based scaffolds, due to natural cells binding via integrin receptors [241]. Additionally, the C4S is able to enhance the metabolic activities of cells [242], which may explain the increased proliferation seen on the COL-C4S scaffolds. By contrast, elastin is less favorable to cell proliferation, probably because of its non-integrin signaling pathway [243], causing the cell proliferation on the COL-EL scaffolds to slow down after day 4 (see Figure 16). These results are in agreement with a previous study [244] that investigated bicomponent construct COL-C4S hydrogels to engineer a mitral valve tissue. The presence of C4S has a positive influence on the bioactivity of seeded cells and also on tissue remodeling. Additionally, immunohistochemical analyses shown an enhanced elastin and laminin expression, along with a vasoactive molecule (endothelial nitric oxide synthase) expression, which is known to regulate the contractile status of smooth muscle and cardiac muscle cells and was absent in collagen only constructs [244]. These results were an important starting point in the design of more complex structure scaffolds, namely a bilayer scaffold from collagen (COL) and elastin (EL), with one collagen-rich layer resembling the fibrosa and one elastin-rich layer resembling the ventricularis [210].

Bilayer scaffolds proved to have anisotropic bending moduli, mimicking the characteristic behavior of the native heart valves. Moreover, an asymmetrical distribution of cardiosphere-drived cells (CDCs) in the two different layers was observed. After 7 days of culture, CDCs show good proliferation on both scaffold layers, but it turns out that CDCs prefer COL instead of EL when proliferating, as found in the previously discussed study [230], where the cause was the presence of different signaling pathways for cellular receptors on COL and EL. In addition, the higher susceptibility of EL to cell contraction strongly influences the contraction of the scaffold, limiting the space for cell proliferation [210]. The evaluation of the COL-EL mixtures was also performed by Wang et al. [231], which designed more complex 3D scaffolds based on COL-EL hydrogels. This time, two types of cells were encapsulated in two different areas of the scaffolds, more exactly the porcine aortic valve interstitial cells (PAVICs) inside the hydrogels and endothelial cells (PAVECs) on their surface, to create an in vitro 3D VEC-VIC co-culture [231]. VICs continuously proliferated up to 7 days, while the cells doubled in number and changed their morphology becoming more elongated and aligned with time. Additionally, VICs exhibited stable levels of β1 integrin and F-actin expressions during the entire culture period (Figure 17). In contrast, although the proliferation of VECs on the gel surface appeared to be very good, cellular losses were observed over time and β1 integrin expression remained low. On day 7, over 20% of VECs transformed into a mesenchymal phenotype, indicated by increased actin filaments and β1 integrin expression (Figure 18). Thus, 3D COL-EL scaffolds proved to possess an environment bio-chemically similar to those of native heart valves, supporting cell attachment, differentiation and proliferation. More importantly, these 3D scaffolds resulted from temperature triggered gelation without chemical crosslinker, which both simplifies the procedures and reduces toxicity [231].

Collagen-chitosan (COL-CH) composite films, with a mass ratio of 7:1 (COL:CH), have proved to be a suitable material for scaffolding, due to their moderate flexibility, high resilience and resistance to tension. Moreover, COL-CH composites formed 3D structures with interconnected pores, and dimensions appropriate for tissue-engineered scaffolds in the range of 80—260 mm. The COL-CH scaffolds were seeded with three types of cells harvested from New Zealand white rabbits, resulting in four groups: *SMC group*—smooth muscle cells (SMCs) and fibroblasts (FIs); *EC group*—endothelial cells (ECs); (*SMC+EC*) *group*—SMCs, fibroblasts and ECs; and *blank control group*—without cells, respectively. Each cell group was seeded twice, waiting 24 h between cell seeding. The first seeding of SMCs and FIs improved the scaffold environment for the further ECs adhesion, differentiation and growth. Moreover, the sequential seeding of SMCs, FIs and then ECs into the valve scaffolds led to significant increases in the secretion of 6-keto-PGF1a compared with scaffolds seeded with ECs alone. Thus, the seeded ECs not only adhere and proliferate, but also secrete vasoactive substances, demonstrating the biological activity of the manufactured composite scaffolds. SEM images, for the *SMC group*, showed a large number of cells inside the scaffolds with dense disordered arrangement, while for the (*SMC+EC) group*, a large number of scattered ECs, with a long shuttle shape were observed. Therefore, the COL-CH composite scaffolds are cytocompatible, supporting the endothelial cell differentiation, have good cell adhesion and biological activity, proving that they may be a useful candidate for tissue engineering materials for artificial heart valves [232].

Hybrid scaffolds from type I collagen and hyaluronic acid (COL-HA) were used to prepare aortic valve extracellular matrix with tailorable crosslinking densities [233]. The crosslinking route was based on N-(3-dimethylaminopropyl)-N’-ethylcarbodiimide hydrochloride/N-hydroxysuccinimide (EDC/NHS) to obtain a heterogeneous microstructure with preserved triple helix structure. Cardiosphere-derived cells (CDCs), selected for seeding on hybrid scaffolds, were isolated from adult Sprague-Dawley rats and cultured for 7 days. Microstructural characterization of the scaffolds has revealed a structure similar to the fibrosa layer in the aortic valve leaflets, resulted by interlaced chains of COL and HA at the microscale. CDCs attachment was mainly affected by the number of available sites not engaged in the crosslinking process and was not affected by the scaffold pore size and stiffness. ECM resulted after the CDCs proliferation and attachment increased the bending moduli of the interlaced scaffolds, but changed the proliferation rate by day 7. This implies that the ECM accumulation altered the surface properties of the scaffolds and the properties of the scaffold do not affect the cell behavior after 1 week. Thus, the CDCs/COL-HA combination has potential to serve as a sustainable cell-material system for valve repair. Moreover, the control of the crosslinking density provides a simple approach to optimize the desirable combination of structural stability and cell-compatibility for any tissue engineering application [233].

In summary, the collagen is of high interest for HVTE, mainly because the leaflets of the valve are primarily composed of collagen type I. It can be obtained by simple decellularization of biological matrices, which results in more porous structures with 3D architecture, suitable for cell adhesion and proliferation. The results presented above prove the high cells adhesion to collagen, due to its natural cells binding via integrin receptors. The cells proliferate on the scaffolds with excellent viability, suggesting that collagen is non-cytotoxic and is inductive for cell growth. Maybe that is why collagen is the most common component of engineered valves. Its only disadvantage, however, remains the weak mechanical properties of the collagen-scaffolds compared with native tissue, suggesting the need for higher crosslinking density or better structural control by combination with other materials, leading to stronger constructions.

#### 5.2.2. Fibrin-Based Scaffolds

Another natural biomaterial with potential in TEHV is fibrin, which is a fibrillar protein and the product of the coagulation pathway involving the blood proteins fibrinogen and thrombin [185]. This natural polymer has a number of advantageous properties, which include its autologous origin, rapid polymerization, adjustable degradation and manufacturability in 3D geometries [32]. Moreover, the fibrin network has a nanometric fibrous structure that mimics the ECM [245], supports the growth and proliferation of cells and allows biologically active molecules to bind to it [185]. However, valves obtained using only fibrin face several problems related to their mechanical properties, their degradation and contraction, due to structural changes and contraction of newly synthesized collagen bundles [234]. Sometimes, some of these critical problems can be solved by introducing fibrin into various composite materials or polymer mixtures [245,246] (see Table 6).

Flanagan and coworkers conducted several studies on developing of TEHV by using fibrin-based materials [145,234]. For instance, a completely autologous fibrin-based heart valve was designed using a molding technique and seeded with ovine carotid artery-derived cells (OCAs). The manufactured fibrin-based heart valves were then subjected to mechanical conditioning in a custom-design bioreactor system, for 12 days (“conditioned valves”) [234], in order to improve the matrix composition and mechanical properties of the tissue construct. The cell phenotype and ECM composition were compared with those of native ovine aortic valve tissue, cultured under agitation in a beaker, called “control valve”. The application of low-pressure conditions and pulsatile flow enhances the seeded cell attachment and alignment within fibrin-based heart valves, but also significantly changes the manner in which these cells generate ECM proteins and remodel the valve matrix. The “conditioned valves” have shown a well-organized fibrous tissue structure with an aligned cell population throughout the entire thickness of the leaflets, while in “control samples”, there are free spaces that surround the cells, suggesting cellular detachment from the scaffold and possible cell death (Figure 19). After 12 days of conditioning, the immunohistochemistry revealed aligned, vimentin-positive cells, suggesting a favorable seeded cell phenotype. Additionally, the synthesis of the glycoproteins, fibronectin and laminin indicates considerable remodeling of the surrounding ECM [234]. Two years later, the same research team [145] achieved in vivo tests by implanting these fibrin-based scaffolds seeded with carotid artery myofibroblasts in sheep. The fibrin-based valves were implanted, for 3 months, in the pulmonary trunk of the same animals from which the cells were harvested. After 3 months in vivo, the fibrin scaffolds had been completely resorbed and replaced by ECM proteins, along with a significant tissue development and cell distribution.

Tubular leaflet design for fibrin-based heart valves attracted the interest of several researchers due to the construction simplicity and the reliability of the implantation technique [120,140,236]. In 2013, Syedain and coworkers [236] manufactured a fibrin-based tubular heart valve and encapsulated with ovine dermal fibroblasts. The tissue tube of 4 mm in diameter was mounted on a frame with three struts, which upon back-pressure, cause the tube to collapse into three coapting “leaflets”. The engineered tissue tubes displayed compositional and mechanical properties similar to those of native ovine heart valve tissue, along with a high degree of mechanical anisotropy. Moreover, the tubular fibrin has proven its excellent performance under hydrodynamic conditions, within a pulse duplicator system, with minimal regurgitant fractions (approx. 5%). Additionally, the transvalvular pressure gradients at the peak systole and effective orifice areas exceeded those of commercially available valves (i.e., bioprosthetic valves from porcine AV leaflets and bovine pericardium). In vivo evaluation of these engineered tubes was made by implantation, as femoral grafts, into sheep and showed a substantial recellularization after 2 months, with no significant change in their diameter and mechanical properties and no observable macroscopic tissue deterioration [236].

Weber et al. [140] proposed a fibrin-based tubular heart valve based on different design, consisting of a simple tubular construct, sutured along a circumferential line at the root and at three single points at the sinotubular junction. In vivo studies, performed in a sheep model, revealed the absence of thrombus formation, calcification, stenosis or aneurysm development. Three months after implantation, a confluent monolayer of endothelial cells was detected on the entire surface of the valve and also an extensive endothelialization on the root, but not on the leaflets. All these excellent results, along with the advantage of a one-piece construct manufacturing method without the need of glue, which could negatively influence the leaflets flexibility or induce calcification, shows the potential of TEHVs based on an autologous fibrin scaffold [140].

A different tubular construct based on fibrin gel was developed by Moreira et al. [120]. Fibrin gel and human umbilical vein cells were molded as tube-in-stent form, and sewn into a self-expandable nitinol stent to forms three coaptating leaflets by collapsing under diastolic back pressure. Tissue analysis by conventional hematoxylin and eosin (H&E) showed homogeneous distribution of cells throughout the valve and also the deposition of collagen fibers oriented along the longitudinal direction. The simulation of the catheter-based delivery involved subjecting the valves to crimping for 20 min. This procedure does not influence the valve’s mechanical properties or its functionality. Thus, by combining tissue engineering with minimally invasive implant technology, a functional fibrin-based heart valve with a simple tubular design can be manufactured [120].

Biomimetic heart valves, with different compositions and different cell lines in the valve wall and leaflets, respectively, were developed in order to mimic the heterogeneity of heart valves [32]. A multi-step injection molding technique was used to mold the valve wall from fibrin gel and the leaflets from fibrin/elastin-like recombinamer (ELR) hybrid gel, respectively. Additionally, ovine umbilical artery cells (OUAs) were seeded in leaflets and ovine carotid arteries cells (OCAs) in the valve wall, respectively. The ELR bearing lysine groups promotes a covalent reaction with the glutamines from fibrin to obtain hybrid gels [247] and the repetition of pentapeptide sequences present in the natural elastin offers the possibility to enhance the elastic properties of the scaffold while being biocompatible and nonimmunogenic [32]. The construct’s cohesion and functionality were demonstrated by opening/closing cycles in a bioreactor and with continuous stimulation over 2 weeks. Immunohistology analysis confirmed the tissue formation and different cell type localization in the leaflets and wall: the vessel-derived α-SMA-negative in the leaflet and α-SMA-positive cells in the wall. The multiple-step molding technique proved to be a versatile tool toward the fabrication of biomimetic TEHVs. The technique is easy to implement and does not require gluing or suturing parts together, which could influence the stiffness of leaflets or lead to calcification points [32]. One year later, Moreira and coworkers developed another type of biomimetic valve based on fibrin gel, known as BioTexValve [134]. The engineered leaflets for the aortic position valve were based on bio-inspired anisotropic composites that combine the biofunctionality of fibrin gel as a cell carrier, and the mechanical strength of synthetic polymers electrospun fibers. The composite leaflets were produced by placing 12 multifilaments of poly(L-lactide-co-D,L-lactide) (PLDL) on a template (Figure 20A) and fixing them with electrospun poly(lactic-co-glycolic acid) (PLGA) fibers (Figure 20B). After the molding process, a complete, resistant and compact valve is obtained due to the presence of multifilament fibers that “embrace” both the leaflets and the valve wall (Figure 20E). The next step was to completely incorporate the composite textile scaffolds into the cell-laden fibrin gel.

BioTexValve showed anisotropic Young’s modulus values comparable with that of the native aortic leaflet, being able to withstand aortic flow and pressure conditions when tested in a flow-loop system. As a consequence of the ECM synthesis, the burst strength increases from 720 to 1086 mmHg; this is approximately ten times higher than the average pressure to which the leaflets are exposed during the diastolic phase (100 mmHg). Immunohistochemical and H&E staining illustrated homogeneous cell distribution and the presence of α-SMA aligned along the longitudinal direction of the wall and leaflet. All these positive results make this fibrin-based composite a potential material to obtain functional tissue-engineered implants with a biomimetic design [134].

Another composite nanofibrous scaffold based on a fibrous protein and electrospun fibers has been reported by Du et al. [237]. In this experiment, the components of the composite were silk fibroin (SF) and L-lysine diisocyanate poly(ester-urethane) urea (LDI-PEUU) fibers obtained by electrospinning. SF derived from silkworm has excellent biological properties, good flexibility and good moisture permeability [248]. On the other hand, LDI-PEUU is an elastomer with good mechanical properties, adjustable processability and good biocompatibility [249]. Thus, the composite will maintain excellent biological properties and acquires improved mechanical properties [237]. SEM and AFM measurements showed a framework constituted by randomly oriented fibers, with a smooth and homogenous surface and a porous 3D structure. The increase of the LDI-PEUU ratio improved the mechanical properties of nanofibers, but at the same time, decreased the degradation rate of composites. SF/LDI-PEUU composite has good blood compatibility, the hemolysis rate being less than 5%. The biocompatibility of the composites was also confirmed by the viability evaluation of the human umbilical vein endothelial cells (HUVECs) in direct contact with the scaffolds for 1, 4 and 7 days, respectively. For instance, on SF scaffolds, the seeded HUVECs are rounded and scattered, but on SF/LDI-PEUU nanofibrous scaffolds, the cells are spindle-shaped and spread well on the scaffolds’ surface. Overall, the above results indicate that the SF/LDI-PEUU composite scaffolds have greater potential for use as heart valve scaffold in tissue engineering. Although the incorporation of LDI-PEUU fibers led to a slight decrease in the SF scaffolds’ hydrophilicity, they improved the mechanical properties and cell proliferation, so that the composites showed more suitable tensile and cell viability properties compared to pure SF and LDI- PEUU fibers [237].

In summary, fibrin can easily be injected into a mold to generate a complex 3D scaffold that promote healthy cellular ECM production and remodeling without resulting in thrombosis or calcification. The major benefit of using fibrin is that fibrinogen and thrombin can be extracted directly from the patient’s blood, and thus, minimizing concerns of immunogenicity [4]. Although fibrin meets the criteria for cell–matrix interactions, there are concerns with regards to the material’s mechanical properties. Thus, a hybrid valve conduit may be the ideal approach for overcoming some of the limitations previously stated [185].

### 5.3. Structure-Properties-Functionality Correlations in HVTE

In the Section 5.1 and Section 5.2 respectively, a series of scaffolds based on the most well-known polysaccharides and proteins have been discussed in detail, with focus mainly on the scaffolds composition and their function in heart valve tissue engineering (HVTE). However, at the same time, the structural properties also affect the functionality of these materials. Thus, in the following, we aim to highlight the direct interdependence that exists between the structural properties of scaffolds and their efficiency in HVTE.

The scaffold, as mentioned earlier in this paper, is one of the most important entities to be considered for efficient tissue engineering, because its properties affect both the generation of the tissue construct in vitro and its post-implantation functionality. Along with the biological and mechanical properties, an important role is played also by the structural properties of the scaffolds, which refer mainly to external geometry, surface properties, pore density, pore size and interconnectivity, interface adherence, etc. [17,18,250].

From Table 5 and Table 6, it can be seen that the scaffolds based on polysaccharides and proteins, respectively, show a wide diversity, both compositional and structural. This is reflected in materials such as simple casting films, electro-spun membranes or fibers and continuing to the most complex hybrid, fibrous, nanostructured or multilayered scaffolds. Starting from this idea, in Table 8, we tried to systematize these scaffolds, obtained by different techniques, and highlight how their structural properties influence their efficiency as scaffolding materials in HVTE.

The native heart valve is a complex trilayered structure consisting of collagen, elastin and glycosaminoglycans; thus, any hydrogel/composite/hybrid-like material that approaches its structure and composition is the most suitable scaffold for engineering a heart valve construct. Over time, various techniques have been used to obtain scaffolds with different structural features, in order to best perform the functions of the heart valve tissue. The conventional fabrication techniques, such as solvent casting to obtain films or membranes [164,165], do not enable the fabrication of complex architectures with precise control of pore size and geometry, pore interconnectivity or spatial distribution within the scaffold. In this case, although materials with a uniform structure, small thickness and good mechanical strength can be obtained, sometimes problems arise due to poor adhesion of the cells to scaffold surface or low spreading of the cells within the scaffold. In this case, it is necessary to immobilize on the surface of the materials clues, such as the adhesive proteins (fibronectin and mouse laminin) [164] or the growth factors (bFGF, basic fibroblast growth factor) that increase the cells adhesion to the surface [165].

With the development of new techniques (i.e., electrospinning, 3D printing, etc.), heterogeneous 3D scaffolds with strong mechanical strength and with the optimum characteristics of an ideal scaffold for cardiac tissue engineering, such as the morphology and accuracy of native ECM, can be fabricated. In this context, the feasibility and attractiveness of the electrospinning technique can be mentioned in the fabrication of porous 3D structures with small thickness, controlled porosity and optimum fibers diameter [167]. Generally, the porous scaffolds are distinguished by an interconnected homogeneous pore network, providing a continuous flow of nutrients and metabolic waste to enable cells growth, proliferation and spreading. Regarding this technique, it also allows obtaining fully aligned or randomly oriented fibrous scaffolds, which positively influence the mechanical properties, pores dimensions and interconnectivity, and implicitly, the proliferation and the distribution of the cells [169,237].

Hydrogels and IPNs hydrogels are complex structures, having ECM-like 3D microstructure and great mechanical strength that support both 2D surface-seeded cell culture and 3D cell encapsulation. They have the advantage of being obtained by a light technique, in different shapes, sizes and thicknesses, with porosity and mechanical strength controlled by changing the degree of crosslinking. Due to the porous 3D network structure, they are characterized by an excellent cell proliferation and extensive cell spreading with a faster migration rate. Their compatibility with biological tissues, high water content and relatively good mechanical properties make these materials attractive for HVTE applications. Moreover, by adding cells to hydrogel before the gelling process, these can be distributed homogeneously throughout the resulting scaffold [41,170,171,175,176].

Fibrous scaffolds are superior scaffolds in term of cell adhesion, migration, proliferation and differentiation, due to the high aspect ratio of fibers, growth factor loading efficiency and sustained release capacity. The development of nanofibers has enhanced the scope for fabricating scaffolds that can potentially mimic the architecture of natural human tissue at the nanometer scale. For HVTE, fibrous scaffolds provide an ideal environment for cells growth and proliferation, leading to 3D structures with porosity, pore size and mechanical characteristics comparable to native heart valves [168].

As a conclusion, each of the above presented scaffolds have advantages and disadvantages. However, the unique properties of the materials used either alone or in combination with other natural or synthetic polymers work for the purpose to develop new heart valves with the ability to repair, reshape and regenerate the cardiac tissue.

## 6. Challenges and Future Outlook

Despite numerous attempts in recent decades, efforts to create a heart valve that has the same specific properties as native heart valves have proven to be extremely difficult. Although there have been various approaches with impressive results, there are substantial limitations that cannot yet be overcome.

The good mechanical strength of mechanical valves, with long-term use and without a rapid surgical reoperation, is well known. However, all of them are associated with a major clinical limitation related to the significant risk for thromboembolic complications and the necessity of anticoagulation therapy throughout life. The bioprosthetic valves do not require the use of anticoagulants, but these have a limited durability because of their structural deterioration, as a result of progressive degenerative calcific. Moreover, serious limitations related to the permanent risk of infections and thrombosis are associated with the currently available heart valves, the glutaraldehyde-fixed bioprosthetic heart valves. One option in this regard is the use of homograft valves, which have a lower thromboembolic risk and a longer durability than glutaraldehyde-fixed tissue valves. However, even in this case there are severe limitations, related to the widespread deficit of organs, which indicates that in the near future, these may not have a major impact on the replacement of the heart valve for a larger population of patients.

A promising alternative in the achievement of heart valves replacements are the bioabsorbable polymers, materials with an unlimited supply potential and which do not require lifelong anticoagulant therapy. The use of natural polymers in HVTE may introduces distinct advantages in this field, due to: (i) the unlimited possibilities to design biomaterials with the required biological and mechanical characteristics, (ii) the ability to modify their surface or structure, in order to improve the conditions for cells adhesion, growth and differentiation, and (iii) the capacity to control their mechanical properties to be similar to those of native valve, by using composite polymeric materials. These easy-to-design biomaterials, which can reproduce the geometry and hemodynamics of natural valves, have brought new hopes in the development of advanced heart valves. First of all, the unlimited availability of these biomaterials is essential, as most current approaches to heart valve tissue engineering are focusing on the in vitro manufacture of autologous cell-based living constructs. In addition, the intrinsic regenerative capacity of them is another essential feature of these materials. It was demonstrated their controllable biodegrade rate that approximates the rate of tissue regeneration, under the culture conditions of interest, and promote cell-biomaterial interactions, cell adhesion and ECM deposition, causing a minimal degree of inflammation or in vivo toxicity. A key theme in designing tissue-engineered scaffolds is to understand and establish the correlations between scaffold properties and biological functions.

Bioresorbable polymer-based heart valves have already been evaluated in the first in-human clinical trial. This study provides important safety and performance data on heart valves, which will make a basis for their broad clinical translation. Generally, the translation of the manufactured heart valve into clinical practice and then commercialization involves various regulatory and ethical limitations. Therefore, substantial efforts are being made to introduce the good manufacturing practices (GMP) and good laboratory practices (GLP) and to develop standardization guidelines by taking into account all specific national guidelines for the classification of tissue-engineered medical products (TEMP). It is also necessary to develop strict preclinical quality control criteria (in vitro and in vivo) before attempting clinical trials.

Taking into account the enormous research effort in the direction of polymer-based valve replacements fabrication and considering all the recent advances in the field, it is expected that in the near future, a new generation of polymeric valves with a real potential to improve the life quality of the patients will be developed.

## Figures and Tables

**Figure 1 biomedicines-10-01095-f001:**
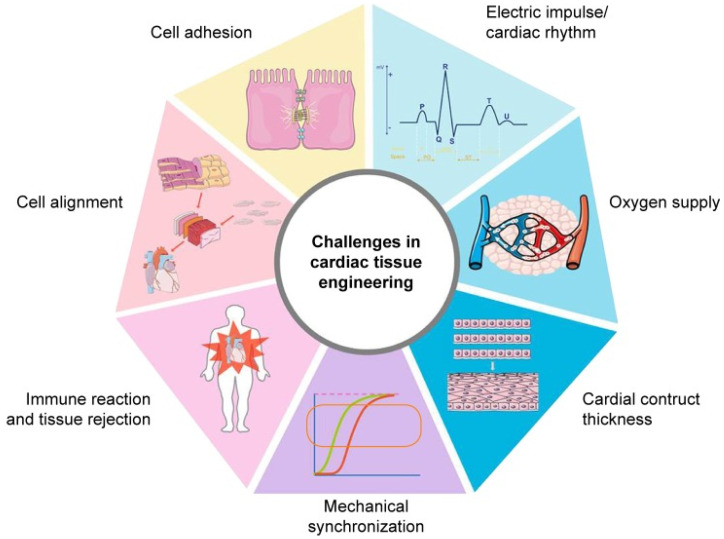
Graphical representation of challenges in cardiac tissue engineering. Reprinted with permission from ref. [35]. Copyright 2018, Dove Medical Press.

**Figure 2 biomedicines-10-01095-f002:**
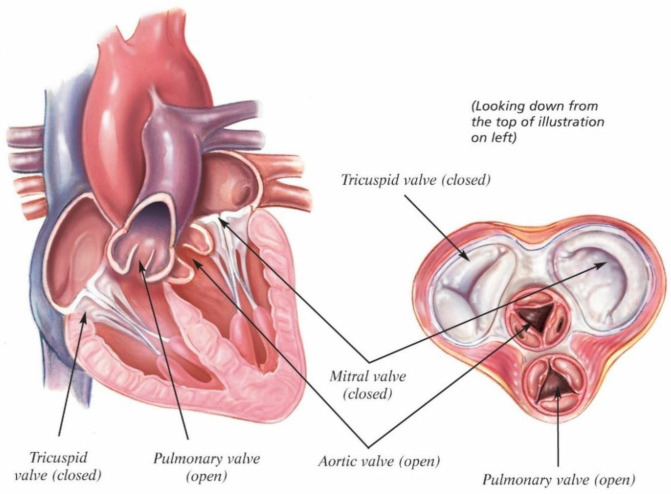
A schematic view of the heart valves. Reprinted with permission from ref. [62]. Copyright 2022, Edwards Lifesciences Corporation.

**Figure 3 biomedicines-10-01095-f003:**
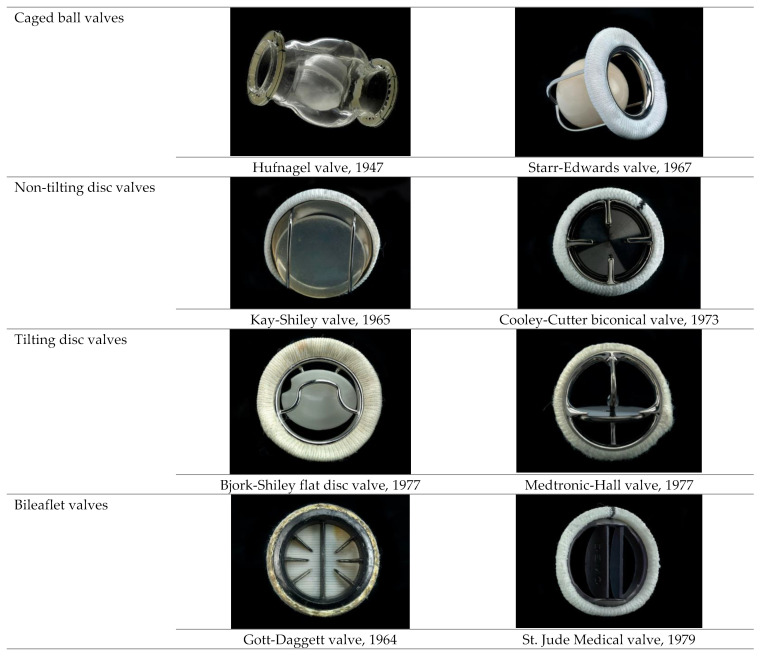
Evolution of mechanical valves. Reprinted with permission from ref. [68]. Copyright 2022, Smithsonian Institution.

**Figure 4 biomedicines-10-01095-f004:**
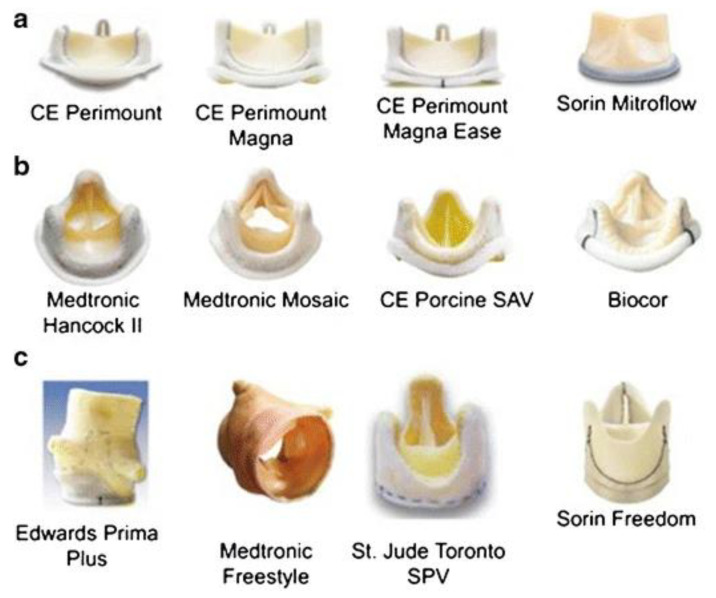
Types of bioprosthetic valves: (**a**) stented pericardial bovine bioprosthetic valves, (**b**) stented porcine aortic valve bioprostheses, (**c**) stentless bioprosthetic valves. Reprinted with permission from ref. [92]. Copyright 2018, Springer Nature.

**Figure 5 biomedicines-10-01095-f005:**
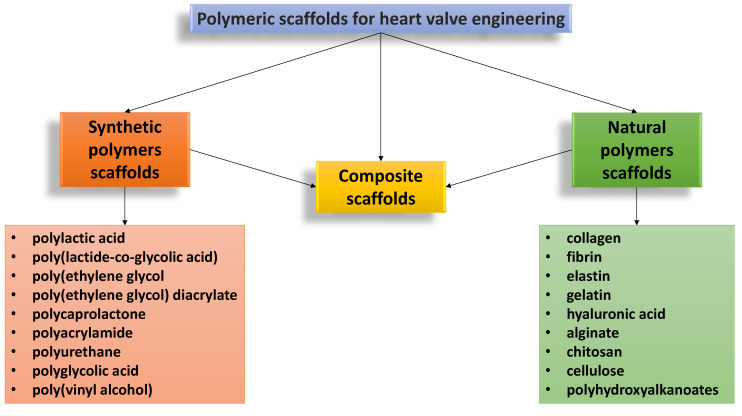
Polymeric scaffolds used in HVTE.

**Figure 6 biomedicines-10-01095-f006:**
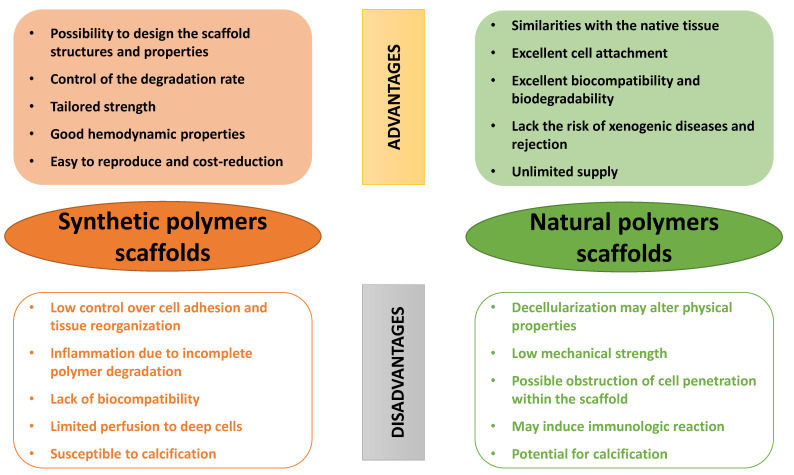
Comparative analysis of the polymeric scaffolds used in HVTE.

**Figure 7 biomedicines-10-01095-f007:**
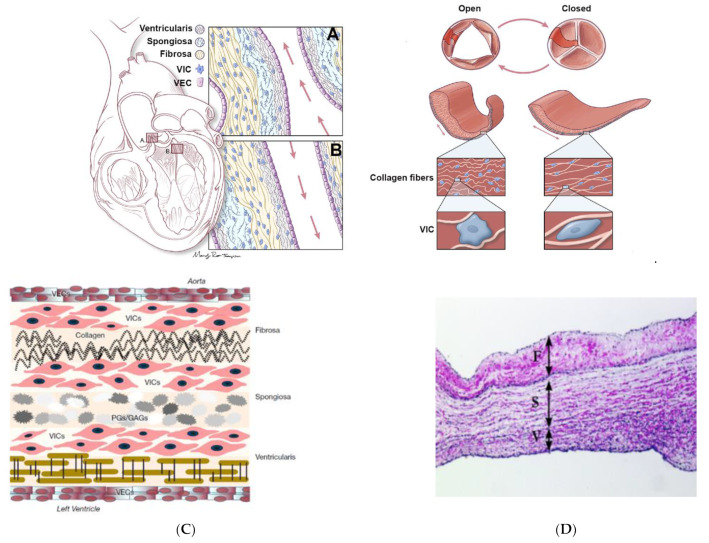
Representation of aortic and mitral valve structures. (**A**) Aortic valve and (**B**) mitral valve, with the three ECM layers: ventricularis (EL), spongiosa (PG-GAG) and fibrosa (COL); the blood flow is indicated by red arrows (ventricularis closest to blood flow); valve endothelial cells (VECs, purple) and valve interstitial cells (VICs, blue). (Right) Representation of the aortic valve indicating coordinated rearrangement of the ECM fibers, and elongation of the VICs during systole (open) and diastole (closed). Reprinted with permission from ref. [111]. Copyright 2020, MDPI. (**C**) Detailed heart valve structure: the three inner layers (ventricularis, spongiosa and fibrosa) with proteoglycans (PG), glycosaminoglycans (GAG), collagen type I and type III, elastin and VICs and the outer layer formed by VECs. Reprinted with permission from ref. [105]. Copyright 2015, Cambridge University Pres. (**D**) Tissue image of trilayered structure of an aortic leaflet in sheep. The three layers consist of fibrosa (F), spongiosa (S) and ventricularis (V). Reprinted with permission from ref. [112]. Copyright 2015, SciDoc Publishers.

**Figure 8 biomedicines-10-01095-f008:**
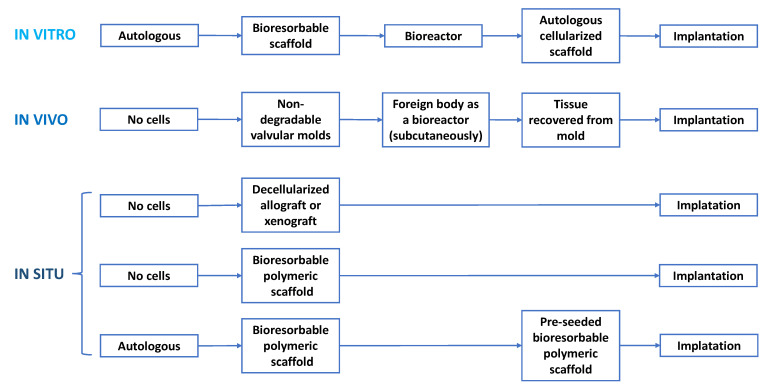
Heart valve tissue engineering strategies: in vitro TEHV, in vivo TEHV and in situ TEHV.

**Figure 9 biomedicines-10-01095-f009:**
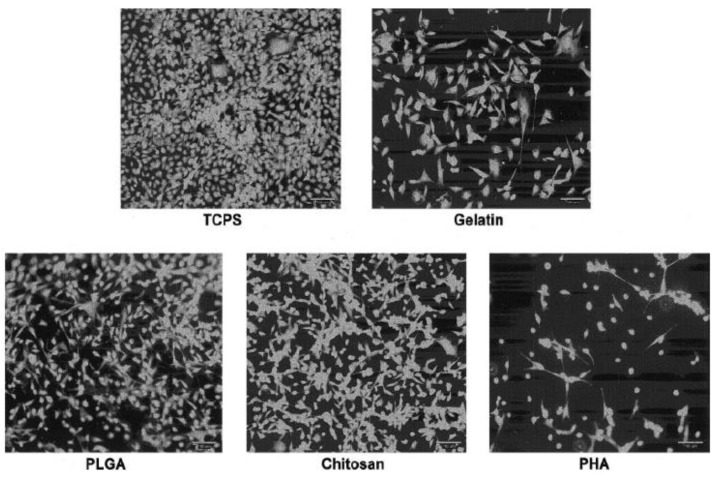
Fluorescence micrographs of VECs seeded onto substrates (bar = 100μm). Reprinted with permission from ref. [164]. Copyright 2003, Wiley Periodicals.

**Figure 10 biomedicines-10-01095-f010:**
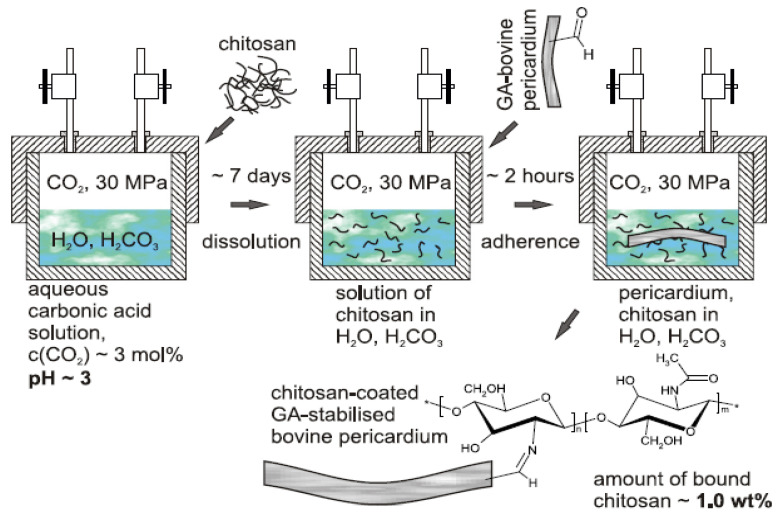
Scheme of bovine pericardium (BP) modification in solutions of CH/H_2_CO_3_. Reprinted with permission from ref. [166]. Copyright 2014, Elsevier.

**Figure 11 biomedicines-10-01095-f011:**
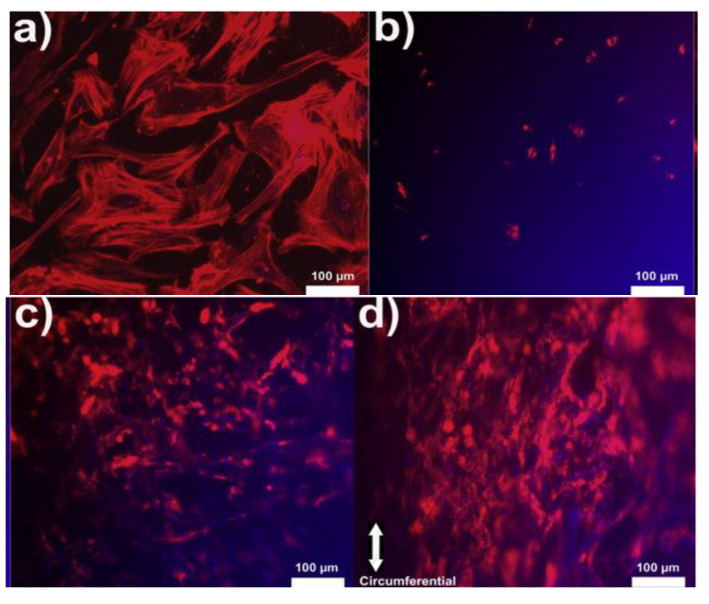
Representative images depicting the cell-scaffold interactions. DiI-DAPI stained hVICs on (**a**) glass coverslip, (**b**) DBP, (**c**) Bio-hybrid R and (**d**) Bio-hybrid A scaffolds; (Dil: 1,1′-dioctadecyl-3,3,3′,3′-tetramethylindocarbocyanine perchlorate; DAPI: 4′,6-diamidino-2-phenylindole). Adapted with permission from ref. [169]. Copyright 2016, Elsevier.

**Figure 12 biomedicines-10-01095-f012:**
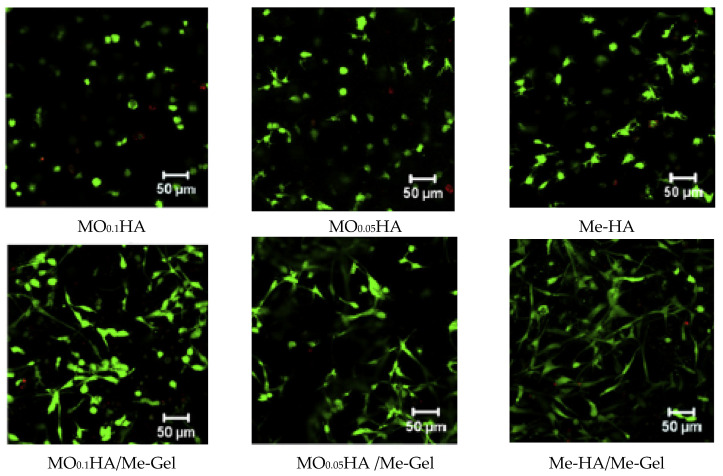
Live/dead assay for encapsulated VICs within HA-based hydrogels, with and without addition of Me-Gel (after 14 days). Adapted with permission from ref. [175]. Copyright 2013, Elsevier.

**Figure 13 biomedicines-10-01095-f013:**
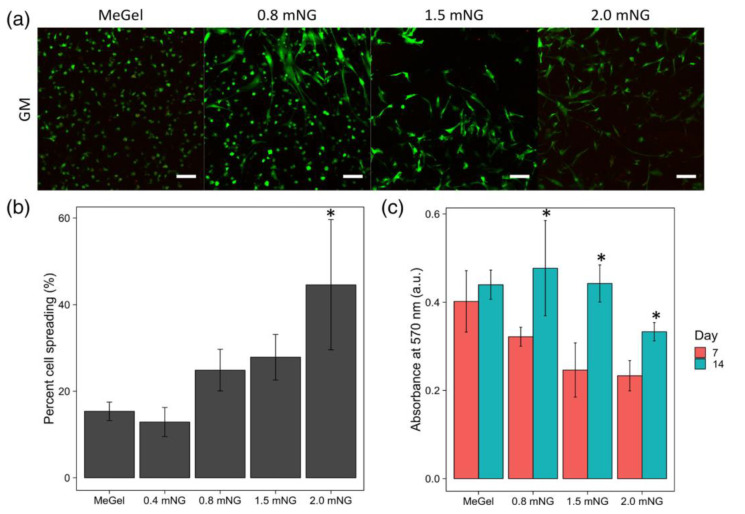
Biocompatibility of Me-Gel and mNG hydrogels. (**a**) Representative images of live/dead staining of cells encapsulated within the different hydrogels (scale bar = 100 μm). (**b**) Percent cell spreading in hydrogels. (**c**) Cell metabolism over time. * *p* < 0.05. Reprinted with permission from ref. [178]. Copyright 2021, Wiley Periodicals.

**Figure 14 biomedicines-10-01095-f014:**
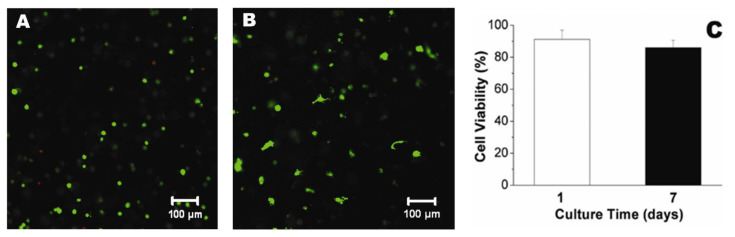
Live/Dead assay for encapsulated VICs within Alg/Gel hydrogel discs, (**A**) after 1 day and (**B**) after 7 days. (**C**) Cell viability measured based on Live/Dead images. Adapted with permission from ref. [133]. Copyright 2012, Wiley Periodicals.

**Figure 15 biomedicines-10-01095-f015:**
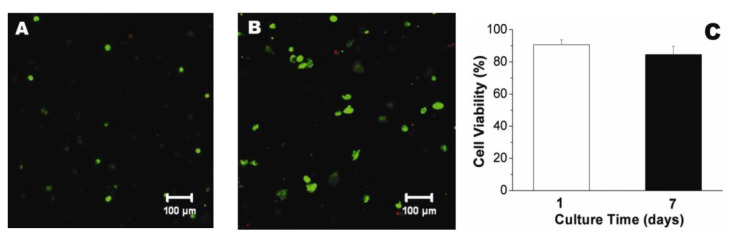
Live/Dead assay for encapsulated SMCs within Alg/Gel hydrogel discs, (**A**) after 1 day and (**B**) after 7 days. (**C**) Cell viability measured based on Live/Dead images. Adapted with permission from ref. [133]. Copyright 2012, Wiley Periodicals.

**Figure 16 biomedicines-10-01095-f016:**
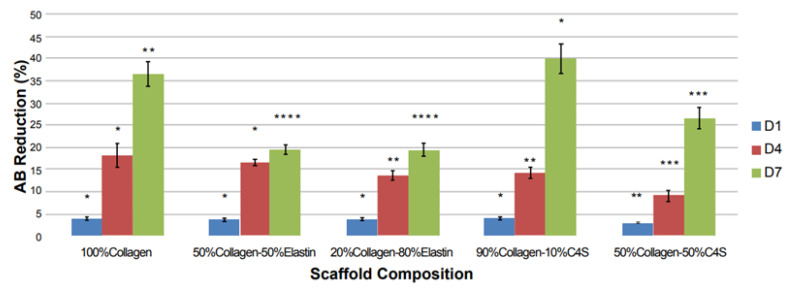
Proliferation of CDCs on scaffolds from day 1 to day 7 (*, ** *** and **** indicate statistically significant differences (*p* < 0.05) between groups of different compositions). Reprinted with permission from ref. [230]. Copyright 2012, Hilaris.

**Figure 17 biomedicines-10-01095-f017:**
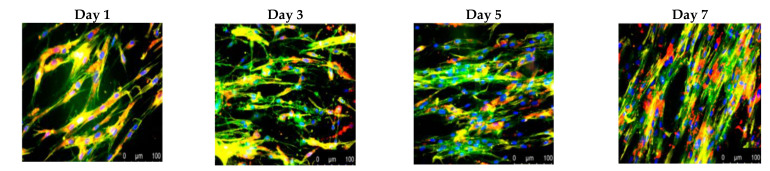
Fluorescent images of VICs in middle layer. Adapted with permission from ref. [231]. Copyright 2018 MDPI.

**Figure 18 biomedicines-10-01095-f018:**

Fluorescent images of VECs on top layer of the VECs-VICs co-culture model β1 Integrin—green; F-actin—yellow; Nuclei—blue; Cell membrane—red (scale bar: 100 μm). Adapted with permission from ref. [231]. Copyright 2018, MDPI.

**Figure 19 biomedicines-10-01095-f019:**
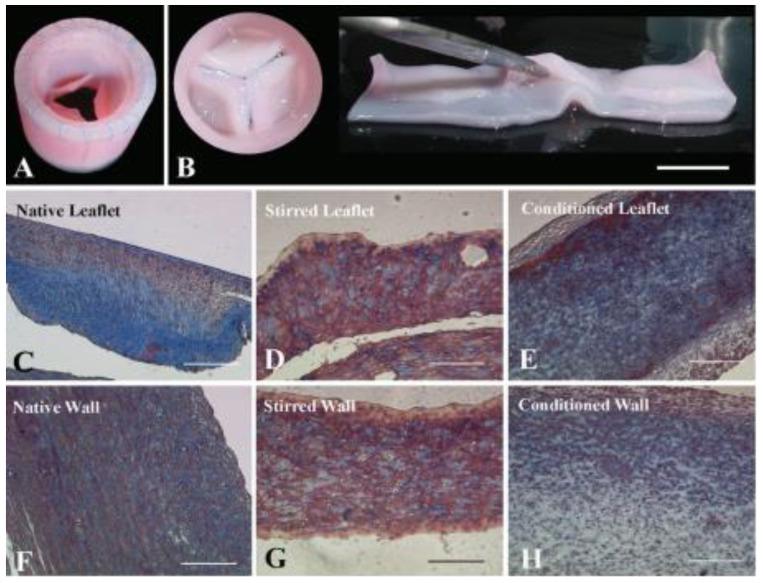
(**A**) Gross appearance of the heart valve after removal from the bioreactor. (**B**) The fibrin-based valves (left: outflow side of closed valve; right: opened conduit cut through conduit wall following removal of the silicone cylinder). Scale: 10 mm. (**C**–**H**) Histological micrographs of trichrome-stained samples: (**C**,**F**) native ovine aortic valve leaflet and aortic wall; (**E**,**H**) conditioned leaflet and wall; (**D**,**G**) stirred tissue samples. Scale: 250 mm. Reprinted with permission from ref. [234]. Copyright 2007, Elsevier.

**Figure 20 biomedicines-10-01095-f020:**
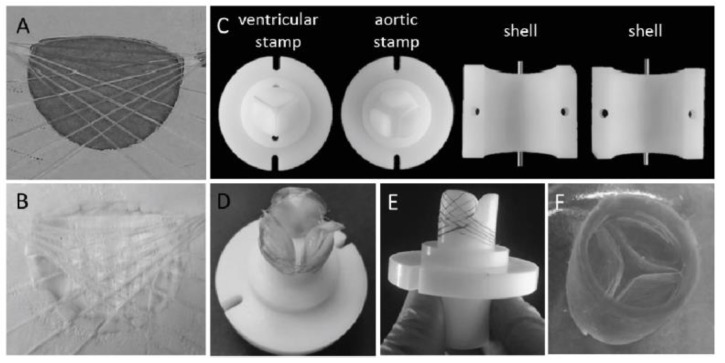
Fabrication process of the BioTexValve. (**A**) Multifilament PLDL fibers fixed on the frame; (**B**) textile composite leaflet after electrospinning; (**C**) molding system; (**D**) leaflets placement on the mold; (**E**) positioning of the PLDL fibers to create continuity between leaflet and wall; (**F**) complete valve fabrication after fibrin gel injection and demolding. Reprinted with permission from ref. [134]. Copyright 2016, Wiley.

**Table 1 biomedicines-10-01095-t001:** Comparative analysis of porous, microsphere, hydrogel and fibrous scaffolds.

Type of Scaffolds	Advantages	Disadvantage
Porous	- tunable mechanical properties; - interconnected homogeneous pore network; - large pore size, which provides continuous flow of nutrients and metabolic waste; - biocompatibility; - enable growth and vascularization of engineered tissues.	- high stiffness and rigidity; - irregular pores size and not interconnected, if no adequate polymeric material is used.
Microsphere	- enhanced structural and mechanical properties; - can impart mechanical support to a weak scaffold matrix - can deliver bioactive molecules in response to environmental stimuli; - act as miniature bioreactors embedded in a surrounding matrix; - serve as cell transporters; - can generate a pores network in the interior of a scaffold to facilitate cellular ingrowth.	- preparation in a multistep process; - residual solvent toxicity; - challenges to control the biomolecule delivery or cell infiltration; - microsphere surfaces with rough appearance depending on the solvent extraction method used; - microsphere size distribution is influenced by the droplet-formation step.
Hydrogel	- structural similarity to ECM; - high water content; - high permeability to oxygen, nutrients and water-soluble metabolites; - mechanical properties controlled by cross-linking of the polymeric components; - tensile strength comparable to that of the aortic valves; - offer high cellular efficiency; - effective cells differentiation, comparable to that of the decellularized scaffolds; - effective materials for bioprinting valve shapes.	- wide distribution of molecular weights and inhomogeneous properties, depending on the preparation method; - risk of side reactions by competing nucleophiles from biological compounds, including living cells; - weak mechanical properties; - lower rigidity by adding cells; - deposited ECM and collagen have no specific orientation since hydrogels do not possess any definite structure.
Fibrous	- high aspect ratio of fibers; - fibrous scaffolds are superior to non-fibrous scaffolds in terms of cell adhesion, migration, proliferation and differentiation; - a high growth factor loading efficiency; - sustained release capacity to specific sites of application.	- biodegradation reduces the mechanical properties of scaffold materials; - the proper material selection is necessary to control the biodegradation rate; - very small pore size of nanofibrous scaffolds can prevents cells from penetrating into the scaffolds.

**Table 2 biomedicines-10-01095-t002:** The classification and the advantages/disadvantages of heart valve replacements.

Types of Valves		Definition	Advantages	Disadvantages
Mechanical valves		- made entirely from metal, pyrolytic carbon and expanded polytetrafluoroethylene (ePTFE or teflon).	- limitless supply; - lack of structural deterioration.	- risk of thrombosis; - requires anticoagulant drugs for life; - not available in small size; - possible mismatch with patients.
Bioprosthetic valves	Autograft valves	- made from another valve within the patient’s own heart (such as the removal of the pulmonary valve to fix the aortic valve).	- not immunogenic; - no risks of thrombosis; - growth potential.	- high probability of replacement after 12 years; - difficult to handle.
	Allograft valves (homograft)	- transplanted within the same species; - from a deceased human donor.	- good hemodynamic profile; - preservation of the morphology; - no risks of thrombosis; - low risk of infection.	- limited availability; - lack of growth potential; - decellularization weakens ECM; - immunogenic response if decellularization not complete.
	Xenograft valves (heterograft)	- transplanted from one species to another - derived from porcine aortic valve or bovine pericardium, implanted in humans.	- limitless supply; - adequate anatomic structure; - optimal biological properties.	- lack of growth potential; - decellularization weakens ECM; - immunogenic response, if decellularization not complete.
Polymeric valves	Natural polymeric scaffolds	- made by cross-linking, photo-polymerization, pressure casting, injection molding, 3D printing, etc.	- limitless supply; - ease of shaping; - polymers combination to meet specific mechanical properties; - combination with stem cells to obtain a living graft.	- degradation by hydrolysis can affect mechanical properties; - possible cytotoxicity of degradation products.
	Synthetic polymeric scaffolds			
	Composite polymeric scaffolds			

Abbreviations: ECM—extracellular matrix; ePTFE—expanded polytetrafluoroethylene.

**Table 3 biomedicines-10-01095-t003:** Overview of cell sources and scaffold materials used for aortic valve replacement.

HVTE	Cell Sources	Study Conditions	Main Results	Ref.
In vitro	Human aortic valve interstitial cells (HAVICs)	- time: 7 days; - scaffold: Me-HA/Me-Gel bioactive hydrogel	- hydrogels’ stiffness regulates the cellular response (the best 4%Me-HA/12%Me-Gel); high cell viability (>90%) for all hydrogels; - GAG deposition markedly higher on day 7 (*p* < 0.01); - the more spreading cells within hydrogels had more expression of genes (α-SMA, vimentin, periostin and collagen I); - the heart valve conduit was successfully printed (4%Me-HA/10%Me-Gel) with acellular root and HAVICs encapsulated leaflets.	[131]
		- time: 14 days; - scaffold: embedded PAN nano-micro fibrous woven fabric into Me-HA/Me-Gel bioactive hydrogel (composite hydrogel)	- HAVICs homogeneously distributed; high cell viability (>90%); improved cell proliferation rate at day 14 (406.4 ± 33.0 ng, *p* < 0.01). - highest levels of α-SMA in hydrogels; collagen content in composite scaffolds lower than hydrogels (*p* < 0.05); - the composite scaffold suppressed transdifferentiation into myofibroblasts and restrains differentiation towards osteoblastic phenotype.	[132]
	HHghHuman aortic root smooth muscle cells (HAoSMCs)	- time: 7 days; - scaffold: alginate/gelatin hydrogels	- viable SMCs encapsulated within alginate/gelatin hydrogel for 7 days; - cell viability after 7 days: 84.6 ± 63.1%; - SMCs expressed α-SMA and vimentin after 7 days.	[133]
	Human umbilical cord vein endothelial cells (HUVECs)	- time: 28 days; - scaffold: fibrin hydrogel in PET warp knitted mesh	- before and after crimping: MPG: 7.3 ± 1.5 mmHg and 6.8 ± 1.7 mmHg; regurgitation: 15.1 ± 2.5% and 15.3 ± 3.6%; - deposition of collagen types I and III orientated along the longitudinal direction; longitudinally aligned α-SMA; - homogeneous cell distribution throughout the valve’s thickness.	[120]
		- time: 21 days; - scaffold: single multifilament PLDL fibers with e-spun PLGA sheet embedded in fibrin hydrogel	- hydrodynamic performance: MPG: 10.7 ± 0.7 mm Hg; the regurgitation fraction: 4.0 ± 1.0%; EOA: 1.4 ± 0.1 cm^2^; - PLDL presence increased the Young’s modulus of the e-spun layer from 2.1 to 7.4 MPa; - α-SMA aligned with the longitudinal direction (wall and leaflet); deposition of collagen types I and III; fibronectin in wall and leaflet.	[134]
	Human umbilical cord blood cells (HUCBs)	- time: 27 days; - scaffold: PGA-P4HB	- good ingrowth of myofibroblasts into the PGA-P4HB scaffolds under cyclic strain; - organized tissue-formation with good ECM formed by myofibroblasts in the inner part of the patches; - collagen: strained 4.06 ± 1.92 mg/mg, perfused 4.21 ± 0.44 mg/mg; GAG: strained 6.44 ± 1.45 mg/mg, patches: 4.65 ± 0.61 mg/mg; cell number was higher in the strained patches.	[135]
		- time: 25 days; - scaffold: P4HB	- cells differentiated into endothelial-like and myofibroblast-like cells; it was formed a confluent monolayer and stained positive for fibroblast, α-SMA and desmin; - the endothelial cell layer, α-SMA (the whole construct); collagen and β1-integrin (leaflets and vascular wall).	[121]
	Human dermal fibroblasts (HDFn) and Ovine dermal fibroblast (ODF)	- time: 24 weeks; - scaffold: fibrin hydrogel	- collagen (24 weeks): 48 ± 8 mg/mL; total protein conc. (24 weeks): 76 ± 14 mg/mL; - cells were positive for an interstitial phenotype (α-SMA and vimentin); laminin and collagen IV (the endothelialized surface); - no evidence of calcification.	[136]
	Porcine aortic valve interstitial cells (PAVICs)	- time: 21 days; - scaffold: PEG-DA with alginate.	- cells viability: 91.3 ± 10.7% (day 1), 100% (day 7 and 21); - viable cells disperse on entire surface; few cells on root and leaflet.	[137]
		- time: 7 days; - scaffold: alginate/gelatin hydrogel.	- viable VICs encapsulated within alginate/gelatin hydrogel for 7 days; - cell viability (day 7): 84.6 ± 63.1%; - VICs expressed α-SMA and vimentin after 7 days; VICs showed higher vimentin expression than α-SMA.	[133]
		- time: 1 month; - scaffold: PCL.	- tensile moduli (aortic valve leaflet): 7.25 ± 2.10 MPa; - VICs demonstrated active fibroblast phenotype (high vimentin, high collagen type I and low α-SMA expression).	[138]
		- time: 28 days; - scaffold: BPUR/PEG.	- heterogeneous distribution of cells; fewer cells along the edges of the scaffold; - compacted cell layers aligned parallel to BPUR; strongly expressed α-SMA and secreted collagen type I; - quiescent VICs growing in PEG; no expression of α-SMA and collagen type I.	[139]
	Ovine umbilical vein endothelial cells (OUVECs)	- time: 21 days; - scaffold: fibrin hydrogel in PET warp-knitted mesh and fibrin hydrogel.	- for PET-fibrin hydrogel, the cells are homogeneously distributed throughout the whole thickness; for fibrin hydrogel, cells appear to be less densely distributed in center; - deposition of collagen types I and III (dynamic conditions).	[140]
	Ovine carotid arteries cells and Ovine umbilical arteries cells	- time: 4 weeks; - scaffold: ELR-fibrin hybrid hydrogel and fibrin hydrogel.	- deposition of collagen I near lumen (TEHV) and homogenous distribution of collagen I in leaflets (native aortic wall); - in TEHV, α-SMA found in conduit wall; α-SMA negative in the leaflet and the density of cells was lower.	[32]
In situ	Ovine dermal fibroblast (oDF)	- time: 4 weeks; - scaffold: fibrin hydrogel.	- collagen aligned circumferentially; - systolic pressure drop: 25 mmHg; EOA: 1.1 cm^2^; - regurgitant fraction: 5% (aortic conditions);	[141]
		- time: 24 weeks; - scaffold: fibrin hydrogel, - implanted in sheep.	- collagen: 48 ± 8 mg/mL (24 weeks); total protein: 76 ± 14 mg/mL; total DNA content: 141 ± 121 mg/mL (24 weeks); - mean systolic pressure drop: (12 weeks) 48 ± 16 mmHg (*n* = 4) and (24 weeks) 45 ± 16 mmHg at (*n* = 3); - measured aorta size (24 weeks): 27 ± 1 mm.	[136]

Abbreviations: BPUR—biodegradable poly(ether ester urethane) urea; DNA—deoxyribonucleic acid; ELR—elastin like recombinamer; EOA—effective orifice area; GAG—glycosaminoglycans; Me-Gel—methacrylate gelatin; Me-HA—methacrylated hyaluronic acid; MPG—mean pressure gradient; P4HB—poly-4-Hydroxybutyric acid; PAN—polyacrylonitrile; PCL—polycaprolactone; PEG—poly(ethylene glycol); PEG-DA—poly(ethylene glycol) diacrylate; PET—polyethylene terephthalate; PGA—polyglycolic acid; PLDL—poly(L/D,L-lactide); PLGA—poly(lactic-co-glycolic acid); SMC—aortic root sinus smooth muscle cells; α-SMA—α-smooth muscle actin.

**Table 4 biomedicines-10-01095-t004:** Overview of cell sources and scaffold materials used for pulmonary valve replacement.

HVTE	Cell Sources	Study Conditions	Main Results	Ref.
In vitro	Human chorionic villous mesenchymal stem cells (CV-MSCs)	- time: 28 days; - scaffold: PGA-P4HB.	- in strained leaflets: GAG: 5.5 ± 0.73 μm/mg; - hydroxyproline: 4.62 ± 2.11 μm/mg; DNA content: 2.78 ± 0.72 μm/mg; - cells expressed vimentin, but lacked α-SMA expression; - after differentiation EPC expressed CD31.	[142]
	Human amniotic fluid cells (H-AFCs)	- time: 28 days; - scaffold: PGA-P4HB.	- immunohistochemistry revealed expression of CD44 and vimentin; - cells started to express eNOS, but no expression of CD31 (28 days); - in strained leaflets: GAG: 12.5 ± 0.81 μg/mg; - hydroxyproline: 2.51 ± 0.79 μg/mg; - DNA content: 4.12 ± 0.77 μg/mg.	[143]
	Cardiac Stem Cells (eCSCs) from Adult Mouse Heart	- time: 15 days; - scaffold: PCL-PLLA nano-micro fiber.	- cells exhibited strong metabolic activities; - stem cells were able to deposited between the fibers.	[144]
	Porcine aortic valve interstitial cells (PAVICs)	- time: 15 days; - scaffold: PCL-PLLA nano-micro fiber.	- improved proliferation with the increase PLLA content; - rotary seeding: better scaffolds penetration of PAVICs; - static seeding: formation of a monolayer of cells.	[144]
	Ovine vascular-derived cells	- time: 28 days; - scaffold: fibrin hydrogel.	- cells expressed vimentin; fewer cells express α-SMA (leaflets and valve wall); cells expressed α-SMA positive (valve conduits). - hydroxyproline: (walls): 20.8 ± 5.5 μg/mg (*n* = 4); - (leaflets): 18.4 ± 6.4 μg/mg (*n* = 4).	[145]
		- time: 20 days; - scaffold: PGA-P4HB.	- regurgitation: 8 ± 3%; orifice area: 1.9 ± 0.1 cm^2^; mean pressure gradient: 5.1 ± 0.3 mmHg; - deposition of collagen: in leaflets and in the outer tissue layers (wall); - DNA: 3.4 ± 0.2 μg/mg; GAG: 12.3 ± 0.8 μg/mg; - hydroxyproline: 9.7 ± 0.6 μg/mg.	[146]
	Ovine bone marrow-derived mesenchymal stromal cells (oBM-MSCs)	- time: 4 weeks; - scaffold: PGA-PLLA.	- tensile strength decreased rapidly in 4 weeks; - diffuse cells expressed of α-SMA throughout the scaffold; cells expressing vimentin evenly distributed.	[147]
		- time: 1 month; - scaffold: PGA-PLLA.	- cusp length decreased: 3.0 ± 0.10 cm (assembly); 2.5 ± 0.16 cm (implantation); - cusp width decreased: 2.8 ± 0.05 cm (assembly); - 1.62 ± 0.23 cm (implantation); valved conduit stable: 2.3 ± 0.07 cm. - cells infiltration in outer layers of scaffold.	[148]
In situ	Human dermal fibroblasts, neonatal (HDFn)	- time: 5 weeks; - scaffold: non-woven PLA mesh; woven Dacron; - implanted in sheep.	- collagen: 24 ± 2 mg/mL (4 weeks), 31 ± 2 mg/mL (8 weeks); elastin: 0.11 mg/mL; - cell concentration: >122% higher compared to implant leaflets (8 weeks); - mean pressure gradient: 2.1 ± 0.8 mm Hg; mean flow velocity: 159 ± 36 cm/s; orifice area: 2.8 ± 0.2 cm^2^.	[149]
		- time: 4 weeks; - scaffold: PGA-P4HB; - implanted in sheep.	- neo-tissue formation and homogeneous ECM deposition; - GAG: 3.65 ± 1.68 μg/mg; hydroxyproline: 18.30 ± 6.34 μg/mg; - deposition of collagen types I and III; - no evidence of regurgitation; no paravalvular leakage.	[150]
	Human vascular-derived cells (vena saphena magna, VSM)	- time: 8 weeks; - scaffold: PGA-P4HB; - implanted in chacma baboons.	- deposition of collagen throughout leaflet and wall; - homogeneous cellular repopulation; - α-SMA positive elements (conduit wall); GAG significantly increased after 8 weeks.	[151]
	Ovine vascular-derived cells	- time: 28 days; - scaffold: fibrin hydrogel; - implanted in sheep.	- deposition of collagen, types I and III; elastin in granular form (leaflets and wall); - hydroxyproline: 28.1 μg/mg (leaflet); 32.7 μg/mg (wall).	[145]
		- time: 8 weeks; - scaffold: PGA-P4HB; - implanted in sheep.	- deposition of collagen in the outer layers; - cells expressed α-SMA and eNOS (wall); cells expressed eNOS and α-SMA (leaflet); cells α-SMA (interstitial); - 8 weeks (% native leaflets): DNA: 86 ± 54%; GAG: 150 ± 11%; hydroxyproline: 26 ± 6%.	[146]
		- time: 24 weeks; - scaffold: PGA-P4HB; - implanted in sheep.	- well-developed ECM; homogeneously repopulated (leaflets and wall); - increased density of collagen (wall and hinge area); lesser extent in the leaflet; - α-SMA negative (leaflet and hinge area); α-SMA positive (wall).	[152]
		- time: 12 months; - scaffold: PGA-P4HB; - implanted in sheep.	- 12 months: mean pressure gradient: 6.1 ± 8.6 mmHg; mean regurgitation fraction: 13.9 ± 5.7%; - increased significantly: collagen: 58.98 μg/mg and DNA content: 1.64 μg/mg; GAG: 18.12 μg/mg (no difference); - substantial amounts of α-SMA positive cells (wall); very few α-SMA positive cells (leaflets); heterogeneous distribution of vimentin-positive cells (wall, hinge, leaflet).	[129]
	Ovine bone marrow-derived mesenchymal stromal cells (oBM-MSCs)	- time: 8 months; - scaffold: PGA-P4HB; - implanted in sheep.	- mean systolic gradient: 9.7 ± 1.3 mmHg; orifice area: 1.35 ± 0.17 cm^2^; - α-SMA positive cells were limited to the subendothelial layer; - 8 months: deposition of collagen (outflow surface); elastin (inflow surface); GAG (valve leaflet).	[147]
		- time: 20 weeks; - scaffold: PGA-PLLA; - implanted in sheep.	- valved conduit diameter stable for > 20 weeks; - density of ECM increased after implantation; - density of cells increased after implantation.	[148]
	Ovine dermal fibroblast (oDF)	- time: 4 weeks; - scaffold: fibrin hydrogel; - implanted in sheep.	- systolic pressure drop: 25 mmHg; EOA: 1.1 cm^2^; - regurgitant fraction: 5% (pulmonary conditions); - fatigue testing (2 weeks): no changes or tissue degeneration (4 weeks); - tissue thinning at contacting with the frame struts.	[141]
		- time: 22 weeks; - scaffold: fibrin hydrogel, - implanted in lamb.	- 22 weeks: deposition of collagen IV (valve root and leaflets); elastin deposition (root and leaflet); - total collagen: 81.2 ± 26.5 mg (explanted root); 49.1 ± 2.04 mg (implant); - total collagen content: 15.2 ± 0.5 mg (leaflets); 4.2 ± 1.8 mg (after implantation); - calcification was not observed (root or leaflets); - 22 weeks: α-SMA expressed near the lumenal surface of the root and partially on the leaflet surfaces.	[153]
	Ovine peripheral vein-derived fibroblasts	- time: 16 weeks; - scaffold: PGA-P4HB; - implanted in sheep.	- newly formed tissue at the interface between the nitinol stent and the native tissue; - collagen content: 60 μg/mg; GAG: 6.4 μg/mg; DNA content: 4.35 μg/mg; - scaffolds demonstrated a high cellular repopulation and ECM remodeling capacity.	[130]

Abbreviations: CD31—clone JC/70A; CD44—clone G44–26; CD44—fluorescein isothiocyanate [FITC]-conjugated (Clone MEM-85); DNA—deoxyribonucleic acid; eNOS—endothelial nitric oxide synthase type III; GAG—glycosaminoglycans; P4HB—poly-4-hydroxybutyric acid; PCL—polycaprolactone; PGA—polyglycolic acid; PLA—polylactic acid; PLLA—poly L-lactic acid; α-SMA—α-smooth muscle actin.

**Table 5 biomedicines-10-01095-t005:** Polysaccharide-based scaffolds for tissue-engineered heart valves.

Scaffold Types	Preparation Methods	Results	Ref.
**Chitosan-Based Scaffolds**			
CH films	Casting method to form films; Adsorption of protein sol. on CH films (4 °C, overnight).	*CH films*: FI/SMCs is less spread and more elongated; *CH/AP*: modest VECs growth, altered elongated morphology, low spreading; *CH/COL IV composites*: enhanced VECs growth, superior cell morphology.	[164]
CH/AP			
CH/COL composites			
(bFGF-CH-P4HB)/DPAV hybrid scaffolds	Coating DPAV with bFGF-CH-P4HB by electrospinning technique (20 kV, room temp.)	bFGF-CH-P4HB fibers form membranes with uniform thickness, firmly attached on DPAV surface; bFGF has a positive effect on the MSCs proliferation.	[165]
CH/BP scaffolds	Immersion of BP tissues in CH/H_2_CO_3_ sol. (pH 3, 2 h, 30 MPa, room temp.)	CH/BP are less rigid and the risk factor of fatigue failureis reduced; Calcification and bacterial strains adhesion are attenuated; In vivo: no inflammatory reaction, after 4 months of implantation in rats.	[166]
CH-PU-GEL nanofibrous scaffolds	Electrospinning technique (16 to 20 kV, room temp.)	OCAs adhered preferentially on CH-GEL-PU, are flattened, spread across the surface and have cobblestone morphology; able to withstand shear-stresses ranging from 0.062 to 0.185 N/m^2^ for up to 3 h;	[167]
CH fibers with immobilized HEP	Extrusion method; HEP immobilization with EDC	Crosslinking degree influences fiber diameters, strength and stiffness; CH-HEP promotes VIC attachment and growth (cell viability ~ 95%, 10 days).	[168]
CH-PCL/DBP biohybrid scaffolds	Electrospinning technique (27–32 °C, 15 kV)	hVICs viability on CH-PCL/DBP (A&R) ~ 90%; Biohybrid (A) has better uniaxial mechanical properties and higher alignment of hVICs compared to a randomly electrospun sample (B).	[169]
**Hyaluronic Acid-Based Scaffolds**			
Me-HA, Me-HA/PEG-DA hydrogels	Photopolymerization (UV light, 5 mW/cm^2^, 3 min, photoinitiator)	Degradation rate: Me-HA/PEG-DA—1 week; Me-HA—2 days; VICs remain viable following photopolymerization; high proliferation after exposure to LMW HA degradation products.	[170]
(Me-HA+CD34)/Me-Gel hydrogels	Photopolymerization (UV light, 180 s, 5.5 mW/cm^2^); CD34 immobilization by EDC/NHS.	Increasing CD34 conc. increases EPC attachment (25.3 ± 5.3 EPCs/mm^2^ at 10 μg/mL; 52.2 ± 5.0 EPCs/mm^2^ at 25 μg/mL); (Me-HA+CD34)/Me-Gel promoted cell elongationand higher spreading.	[171]
SilylHA-CTA/LLDPE IPNs	Silylation of HA-CTA; LLDPE films swollen in silylHA-CTA/xylene (50 °C/1 h).	HA/LLDPE exhibit lower contact angles and less blood clotting than LLDPE alone, which led to considerable thrombus formation; PHVs showed acceptable values for RF (4.77 ± 0.42%) and EOA (2.34 ± 0.5 cm^2^).	[172,173]
HA-LLDPE IPNs/CoCr-MP35N stent	Swelling process was used to obtain IPNs; fixing by PP sutures on the stent frame.	Hemodynamic parameters (EOA, RF, PI) have values comparable with those of commercial transcatheter valves; Turbulent flow tests show a decrease of RSS at each cardiac phase.	[174]
Me-HA/Me-Gel MOHA/Me-Gel hybridhydrogels	Molding technique and exposure to UV light (2 mW/cm^2^; 5 min)	Me-Gel stimulates VICs spread and migration from spheroids; Cell circularity was much lower in low stiffness hydrogels than in stiffer ones; VICs have a spindle-like morphology only in hydrogels with Me-Gel.	[175]
Me-HA/Me-Gel/PGS-PCL hybrid hydrogels	Immersion of electrospun PGS-PCL into hydrogel; Photocrosslinking (UV light, 45 s, 2.6 mW/cm^2^).	MVICs have an initial rounded shape and low spread; MVICs are predominantly spread over the surface of PGS-PCL fibers only; 21 days: MVICs spread is complete into hybrid hydrogels, with non-homogenous distribution at different depths.	[176]
**Cellulose-Based Scaffolds**			
CA coatings for metallic valves	Electrospinning technique; Surface functionalization with RGD and YIGSRG	CA coatings promote cardiac cell growth on valve surface; CA ensures the control of endothelialization and reduction of thrombosis.	[12]
CNF/PU films nanocomposites	Film-stacking method; Compression molding	Prosthetic valves have good biological durability, fatigue resistance and hemodynamics properties; no failure is registered after accelerated fatigue tests, equivalent of 12-year cycles.	[177]
mNG composite hydrogels	Covalent conjugation of mNCC on Me-Gel backbone via NHS/EDC crosslinking	Encapsulated HADMSCs on mNG displayed phenotypic properties found within the heart valve spongiosa; lower expression of osteogenic genes indicates resistance toward calcification.	[178]
BC/PVA anisotropic nanocomposites	Physical crosslinking by freeze-thaw cycles (20 °C/−20 °C); molding technique	Mechanical properties are similar to valve leaflet tissues, in both principal directions; the composition and number of freeze-thaw cycles substantially influence the tissue properties.	[179,180]
	Thermal processing; molding technique	Trileaflet mechanical heart valve mimics the non-linear mechanical properties and anisotropic behavior of the porcine heart valves.	[181,182]
**Alginate-Based Scaffolds**			
PEG-DA/Alg hydrogels	Simultaneous 3D printing/photocrosslink ingmethods	The scaffolds with 10% Alg allow PAVICs to grow along the conduits surface, but less on the root and leaflet interstitium; high cell viability: 91.3 ± 10.7% (day 1) and 100% (day 7 and 21).	[137]
Alg/GEL hydrogels	3D bioprinting with mold extrusion technique	Printing accuracy: 84.3 ± 10.9%; Cell viability (7 days): 81.4 ± 3.4% (SMCs); 83.2 ± 4.0% (VICs); SMCs express α-SMA in stiff matrix; VICs express VIM in soft matrix.	[133]
Dop-Alg hydrogel coatings	Covalent bonding of Dop to Alg (EDC/NHS route); Crosslinking with GA	In vitro: only Dop-Alg determines a decrease in the Ca content: 2.919 ± 0.252 mg/L—day 3; 0.725 ± 0.012 mg/L—day 6; In vivo: the largest decrease in Ca content for Dop-Alg: 1.737 ± 0.124 mg/L—day 20; 0.675 ± 0.084 mg/L—day 30.	[183]

Abbreviations: A&R—aligned and random; ACAN—aggrecan; Alg—alginate; AP—adhesive proteins; BC—bacterial cellulose; bFGF—basic fibroblast growth factor; BP—bovine pericardium; CD34—mouse antibody; Ca—calcium; CA—cellulose acetate; CH—chitosan; CNF—cellulose nanofibrils; COL IV—collagen type IV; CTA—cetyltrimethylammonium bromide; DBP—decellularized bovine pericardium; Dop—dopamine; DPAV—decellularized porcine aortic valve; EC—endothelial cells; EDC—1-ethyl-3-(3-dimethylaminopropyl)-carbodiimide; EOA—effective orifice area; EPC—endothelial progenitor cells; FI—fibroblast; Gel—gelatin; GA—glutaraldehyde; HADMSCs—Human adipose-derived mesenchymal stem cells; HEP—heparin; IPNs—interpenetrated networks; Me-GEL—methacrylated gelatin; Me-HA—methacrylated HA; mNG—mNCC—TEMPO-modified nanocrystalline cellulose; MOHA—methacrylated oxidized HA; MSCs—mesenchymal stem cells; MVICs—mitral valve interstitial cells; NHS—N-hydroxysuccinimide; OCAs—ovine carotid arteries cells; P4HB—poly-4-hydroxybutyrate; PAVICs—porcine aorta valve interstitial cells; PCL—polycaprolactone; PEG-DA—poly(ethylene glycol) diacrylate; PGS-PCL—poly(glycerol sebacate)-polycaprolactone; PHA—polyhydroxyalkanoates; PHVs—polymeric heart valves; PI—pinwheeling index; PP—polypropylene; PU—polyurethane; PVA—poly(vinyl alcohol); RF—regurgitant fraction; RGD—Arginine-Glycine-Aspartate peptides; RSS—Reynolds Shear Stress; SilylHA—silylated HA; α-SMA—α-smooth muscle actin; SMCs—smooth muscle cells; TCPS—tissue culture polystyrene; VICs—valvular interstitial cells; VIM—vimentin; YIGSRG—tyrosine-isoleucine-glycine-serine-arginine-glycine malinins.

**Table 6 biomedicines-10-01095-t006:** Protein-based scaffolds in heart valve tissue engineering.

Scaffold Types	Preparation Methods	Results	Ref.
**Collagen-Based Scaffolds**			
3D-COL biological scaffolds	Decellularization by SDS extraction; crosslinking (EDC/NSH); enzymatic treatment to remove elastic fibers.	Mechanical properties of 3D-COL controlled by crosslinking degree; 3T3 cells adhere and proliferate on COL scaffolds and infiltrated to depth of about 20 mm after 7 days, and 40 mm after 28 days.	[227]
COL/NRASMC matrix	Collagen-cell suspension was cast into silicon rubber wells and cultured in an incubator.	Uniform tension, during COL compaction, increases the cell content, stimulates their metabolism and leads to stronger constructs; NRASMCs are metabolically active proved by the elastin inside and around the COL fibers, and the proteoglycans at their interface.	[228]
3D COL disc scaffolds	Molding technique by rapid prototyping with 3D inkjet printer	VICs proliferate more on 1% *w*/*v* COL than 2% or 5%; VICs remodel the scaffold and synthesize new matrix (detection of remodeling enzymes, MMPs and ECM gene expression).	[229]
COL-EL; COL-C4S heterogenous scaffolds	Molding technique using PTFE molds, followed by freeze drying	Good cell proliferation on COL, due to natural cells binding via integrin receptors; C4S increase the cell metabolic activities; Low cell proliferation on EL, due to its non-integrinsignaling pathway.	[230]
COL-EL bilayer scaffolds	Solution casting into PTFE molds; freeze drying, repeated twice to obtain bilayer structure.	Bilayer scaffolds have anisotropic bending moduli similar to native valves CDCs prefer COL over EL when proliferating, resulting in asymmetrical cell distribution in the two different layers.	[210]
3D COL-EL hydrogels scaffolds	COL-EL composition: 50% COL, 12% EL, 10% PBS, 28% equal parts of DMEM and FBS; pH 7.5; 37°C; 1 h.	3D COL-EL scaffolds support cell attachment, proliferation and differentiation: after 7 days, *VICs* double their number and exhibited stable levels of integrin β1 and F-actin expression; *VECs* have a very good proliferation, but the integrin β1 expression remained low.	[231]
3D COL-CH composites scaffolds	COL:CH (7:1, *w*/*w*) The composites were seeded with 3 types of cells: SMCs, FIs and ECs.	3D COL-CH have good cells adhesion and support ECs differentiation; *SMCs group*—large number of SMCs with dense disordered arrangement; *SMC+EC group:* large number of scattered ECs with long shuttle shape.	[232]
COL-HA hybrid scaffolds	Crosslinking by EDC/NHS route.	Structure similar to fibrosa layer of the valve leaflets; CDCs attachment not affected by the pore size and stiffness.	[233]
**Fibrin-Based Scaffolds**			
Autologous fibrin-based heart valve scaffolds	Molding technique; In vitro: bioreactor conditioning; In vivo: implantation in sheep pulmonary trunk (3 months).	In vitro: well-organized structure of “conditioned samples”, aligned OCAs in leaflets; cellular detachment, possible cells death in “control samples”; In vivo: fibrin scaffolds completely resorbed and replaced by ECM proteins; significant tissue development and cell distribution.	[145,234]
(SC-F) composites biological valves	Coating DPPV with stem cells-fibrin complex	*Static condition*: 1st day—homogenous distribution of SC; 16th day—cell colony formation in SC-F compared to control (no cell clusters); *Dynamic conditions*: starting with the 4th day, floating composite clots at the inner surface of the valve and leaflets are observed.	[235]
Fibrin-based tubular heart valves	The tube mounted on a frame with three struts which, upon back-pressure, cause the tube to collapse into three coating “leaflets”.	In vitro: excellent performance under hydrodynamic conditions, minimal RF (approx. 5%), excellent values for TGV and EOA; In vivo (sheep, 2 months): substantial recellularization and no significant change in diameter or mechanical properties.	[236]
	Tubular construct sutured at the root circumferential line and at three single points of sinotubular junction.	Advantage of one-piece construct manufacturing method without glue; In vivo (sheep, 3 months): no thrombus formation, calcification or stenosis; formation of ECs confluent monolayer on the valve surface.	[140]
Fibrin-based tube-in-stent heart valves	Fibrin gel and HUVCs molded as tube-in-stent form and sewn into a self-expandable nitinol stent.	Homogeneous cells distribution throughout the valve; The simulation of the catheter-based delivery (the valves crimping for 20 min) does not influence the valve mechanical properties or functionality.	[120]
F-ELR biomimetic heart valves	Multi-step injection molding: the valve wall obtained from F gel and the leaflets from F-ELR gel.	Good structure cohesion and functionality (opening/closing cycles); Different cell type localization: the vessel-derived α-SMA negative (leaflets) and α-SMA positive cells (valve wall).	[32]
F/PLDL-PLGA anisotropic composites BioTexValve	Molding of PLDL multifilaments and electrospun PLGA fibers incorporated within fibrin gel.	Anisotropic Young’s moduli comparable with the native aortic leaflets; The valve withstands aortic flow/pressure conditions in flow-loop system; Homogeneous distribution of α-SMA, aligned with the longitudinal direction of the wall and leaflets.	[134]
SF/LDI-PEUU nanofibrous scaffolds	SF and LDI-PEUU prepared by electrospinning process.	Smooth and porous 3D structure of SF/LDI-PEUU scaffolds with randomly oriented fibers; Good blood compatibility (hemolysis rate <5%); HUVECs have spindle-shaped morphology and good spread.	[237]

Abbreviations: C4S—chondroitin-4-sulfate; CDCs—cardiosphere-derived cells; CH—chitosan; DMEM—Dulbecco’s modified Eagle medium; DPPV—decellularized porcine pulmonary valve; ECs—endothelial cells; EDC—1-ethyl-3-(3-dimethylaminopropyl) carbodiimide-hydrochloride; EL—elastin; ELR—elastin-like recombinamer; EOA—effective orifice areas; F—fibrin; FBS—fetal bovine serum; FIs—fibroblasts; HA—hyaluronic acid; HUVECs—human umbilical vein endothelial cells; HUVCs—human umbilical vein cells; LDI-PEUU—L-lysine diisocyanate poly(ester-urethane)urea; MMPs—matrix metalloproteinases; NRASMCs—neonatal rat aortic smooth muscle cells; NSH—N-hydroxysuccinimide; OCAs—ovine carotid artery-derived cells; PBS—phosphate buffered saline; PLDL—poly(L/D,L-lactide); PLGA—poly(lactic-co-glycolic acid); PTFE—polytetrafluoroethylene; RF—regurgitant fractions; SCs—stem cells; SDS—sodium dodecyl sulfate; SF—silk fibrinoin; α-SMA—α-smooth muscle actin; SMCs—smooth muscle cells; 3T3—mouse fibroblasts cells; TVG—transvalvular pressure gradients; VECs—valvular endothelial cells; VICs—valvular interstitial cells.

**Table 7 biomedicines-10-01095-t007:** Scaffolds pore size and percentage of AlamarBlue reduction of D4 and D7 normalized to values of D1. Adapted with permission from ref. [230]. Copyright 2012, Hilaris.

Composition	Pore Size (μm)	D4/D1	D7/D1
100% COL	147.6 ± 38.4	4.63 ± 0.32	9.39 ± 0.86
50% COL + 50% EL	179.9 ± 35.8	4.70 ± 0.39	5.53 ± 0.69
20% COL + 80% EL	187.6 ± 36.5	3.68 ± 0.30	5.24 ± 0.36
90% COL + 10% C4S	115.6 ± 27.6	3.64 ± 0.15	10.29 ± 0.63
50% COL + 50% C4S	116.8 ± 17.8	3.10 ± 0.19	9.15 ± 0.95

**Table 8 biomedicines-10-01095-t008:** Structural properties of the scaffolds and their efficiency in HVTE.

Scaffold type	Specific Structural Properties	Functionality as TEHV Scaffold	Ref.
Films/membranes	- Simple compact architectures; - Controllable thickness; - Size and geometry of the pores cannot be controlled; - Lack of pore interconnectivity.	- Low cell spreading due to reduced pore interconnectivity; - Poor cells adhesiveness to the surface, depending on the polymer nature; - Necessity to modify the surface to increase its functionality (i.e., adhesive proteins) or loading with growth factor (bFGF) to enhance the cell proliferation.	[164,165]
Extruded fibrous scaffolds	- Fibers’ diameter controlled by extrusion parameters; - Small fiber diameter leads to scaffolds with high strength and stiffness.	- High strength properties that match the appropriate mechanical strength for HV tissue; - Poor cells adhesiveness of the surface, which can be improved with covalently immobilize growth factors and adhesive proteins.	[168]
Electrospun nanofibrous scaffolds	- Porous 3D structures with small thickness, controlled porosity and optimum fibers diameter; - Fully aligned or randomly oriented fibrous scaffolds can be obtained.	- Good cells adherence to substrate and higher spreading; good blood compatibility; - Ability to withstand to shear-stress; - Superior uniaxial mechanical properties of aligned compared to non-aligned (random) fibrous scaffolds; - High cells alignment along the align fibers compared to randomly electrospun samples.	[167,169,237]
Hydrogels/Hybrid hydrogels	- Porous 3D network structures with pores interconnectivity; - Facile fabrication of complex shapes, such a tri-leaflet structure; - Tunable porosity, mechanical stiffness and swelling ratio by varying the crosslinking degree; - Appropriate microenvironment that mimics native ECM.	- Support both 2D surface-seeded cell culture and 3D cell encapsulation; - Excellent cells attachment to hydrogels substrate and good cells spreading and elongation; - Significant difference in cells proliferation rate and spreading in hydrogels with different stiffnesses: increased cell proliferation and extensive cell spreading with a faster migration rate in softer hydrogels compared with stiffer ones.	[170,171,175,176]
Nanocomposites/Anisotropic nanocomposites	- Heterogenous, anisotropic structures, with non-linear mechanical properties; - Custom-designed materials with different composition and broad range of mechanical properties.	- Mechanical strength similar to natural valves leaflet tissue; great stress-strain properties in both the circumferential and axial direction; - Mimic the anisotropic behavior of native HV in both the closing and opening phases of the cardiac cycle; - Non-homogenously distribution of cells at different depths of structures.	[178,181,182]
IPNs hydrogels	- ECM-like 3D microstructure; - Mechanical strength greater than their constituent components; - Tunable stiffness and load-bearing capacity by varying the concentration of each component; - Controllable porosity and viscoelastic properties.	- Bending stiffness almost similar with natural valve leaflets, which withstand physiological forces; - Excellent hemodynamics in a pulsatile flow loop system: RF and EOA parameters values are almost similar to those of natural HV; - IPNs microarchitecture permits cell encapsulation and promotes cell proliferation, spreading and differentiation.	[172,173]
Polymer-bioprosthetic valves composites	- 3D structure that combines the polymers valuable properties (i.e., biocompatibility, hydrophilicity, ability to support cell development, etc.) with the native-like valve performance.	- Long-term durability and mechanical stability; reduced calcification; enhanced hemocompatibility; - Absence of any foreign-body immune response after implantation; lower risk of blood clot formation; - Significant improvements in cell proliferation and homogenous distribution.	[166,174,235]

Abbreviations: bFGF—basic fibroblast growth factor; ECM—extracellular matrix; EOA—effective orifice area; HV—heart valve; IPNs—interpenetrated networks; RF—regurgitation fraction.

## Data Availability

Not applicable.

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
