# Peer review of "Natural Polymers in Heart Valve Tissue Engineering: Strategies, Advances and Challenges"

_biomedicines, 2022, doi:10.3390/biomedicines10051095_

Round 1

Reviewer 1 Report

This is a very comprehensive review. However, the content on polymers is not much covered. A structure-property guideline that affects the function of these materials in Tissue engineering will be helpful.

Author Response

Manuscript biomedicines-1690795

Natural polymers in heart valve tissue engineering: Strategies, advances and challenges

by Diana E. Ciolacu, Raluca Nicu and Florin Ciolacu

Response to Reviewer 1 comments

First of all, we would like to thank to the reviewers for their valuable comments and advices, which helped us to improve our manuscript. Our responses are provided below, and modifications in the text are shown in red color.

Q1. This is a very comprehensive review. However, the content on polymers is not much covered.

A1. Indeed, the range of natural polymers used in HVTE is wide and to avoid loading the manuscript (which is quite large) with too much data, we set out to discuss only the natural polymers (unmodified, chemical modified or in combination with other polymers) which have shown the most significant improvements in the recent years.

Q2. A structure-property guideline that affects the function of these materials in Tissue engineering will be helpful.

A2. We improve the manuscript with a new section (5.3. Structure-properties-functionality correlations in HVTE), where the requested correlation between structure-properties-function of the materials is presented.

Reviewer 2 Report

In this review, the authors aimed to provides a synthesis of the tissue engineered heart valve (TEHV) studies, emphasizing the principles, the recent advancements, the current challenges and the future directions in this field.

Starting from the basic principles of tissue engineering, the advantages and limitations of the scaffolds are emphasized. Different techniques for heart valve replacements fabrication, as well as the evolution of in vitro, in vivo and in situ strategies for tissue engineering applications are also discussed. The complex and dynamic structural components that are needed to accomplish normal heart valve function and the required steps to design and validate novel valve are described, particularly focusing on the natural polymers used in the last years in heart valve tissue engineering.

The study covers some issues that have been overlooked in other similar topics. The structure of the manuscript appears adequate and well divided in the sections. Moreover, the study is easy to follow, but few issues should be improved. Some of the comments that would improve the overall quality of the study are:

  1. Authors must pay attention to the technical terms acronyms they used in the text. Please better stated the aim of the study in the abstract section.
  2. English language needs to be revised.
  3. Limitations of the study needs to be added.
  4. Conclusion Section: This paragraph required a general revision to eliminate redundant sentences and to add some "take-home message".

Author Response

Manuscript biomedicines-1690795

Natural polymers in heart valve tissue engineering: Strategies, advances and challenges

by Diana E. Ciolacu, Raluca Nicu and Florin Ciolacu

Response to Reviewer 1 comments

First of all, we would like to thank to the reviewers for their valuable comments and advices, which helped us to improve our manuscript. Our responses are provided below, and modifications in the text are shown in red color.

In this review, the authors aimed to provides a synthesis of the tissue engineered heart valve (TEHV) studies, emphasizing the principles, the recent advancements, the current challenges and the future directions in this field.

Starting from the basic principles of tissue engineering, the advantages and limitations of the scaffolds are emphasized. Different techniques for heart valve replacements fabrication, as well as the evolution of in vitro, in vivo and in situ strategies for tissue engineering applications are also discussed. The complex and dynamic structural components that are needed to accomplish normal heart valve function and the required steps to design and validate novel valve are described, particularly focusing on the natural polymers used in the last years in heart valve tissue engineering.

The study covers some issues that have been overlooked in other similar topics. The structure of the manuscript appears adequate and well divided in the sections. Moreover, the study is easy to follow, but few issues should be improved. Some of the comments that would improve the overall quality of the study are:

Q1. Authors must pay attention to the technical terms acronyms they used in the text.

A1. Thank you for your observation! All the abbreviations have been checked and corrected.

Q2. Please better stated the aim of the study in the abstract section.

A2. The Abstract was revised and the aim of the review was better stated, as follow:

“Starting from this idea, our review presents a comprehensive overview related not only to the structural components of the heart valve, such as cells sources, potential materials and scaffolds fabrication, but also the advances in the development of heart valve replacements. The focus of the review is the recent achievements concerning the utilization of natural polymers in TEHV, thus an extensive presentation of these is provided, emphasizing the natural polymers used (polysaccharides and proteins), the technological progress with its challenges, as well as their clinical evaluations. The available strategies to design, validate and remodel heart valves are discussed in depth by a comparative analysis of in vitro, in vivo (pre-clinical models) and in situ (clinical translation) tissue engineering studies”.

Q3. English language needs to be revised.

Q3. The language corrections have been made by an English language specialist.

Q4. Limitations of the study needs to be added.

A4. The limitations of the study were introduced in a new section, section 6 “Challenges and future outlook”.

Q5. Conclusion Section: This paragraph required a general revision to eliminate redundant sentences and to add some "take-home message".

A5. The Conclusion section were completely revised and the title was changed in “Challenges and future outlook”.

Reviewer 3 Report

  • What is the difference between HVTE and TEHV in the function and concept?
  • What are the main factors in positively answering regarding cell differentiation in Decellularized heart valves?
  • Prepare a table and compare the advantages and disadvantages of porous, microsphere, hydrogel, and fibrous scaffolds.
  • Page 7/57: this section is not necessary and can be deleted.” The basic principles of cell culture for heart valve tissue engineering commonly involve the following steps”.
  • There are some abbreviations in the tables that require to define first such as in table 1: ECM, table 2: SMA, SMC, MPG, PLDL, …. Please check for all the tables.
  • The authors can use the following reference in this manuscript:

Sabbagh, F., Muhamad, I. I., PaLe, N., & Hashim, Z. (2018). Strategies in Improving Properties of Cellulose-Based Hydrogels for Smart Applications. Cellulose-Based Superabsorbent Hydrogels. Springer International Publishing, 887-908.

Author Response

Manuscript biomedicines-1690795

Natural polymers in heart valve tissue engineering: Strategies, advances and challenges

by Diana E. Ciolacu, Raluca Nicu and Florin Ciolacu

Response to Reviewer 3 comments

First of all, we would like to thank to the reviewers for their valuable comments and advices, which helped us to improve our manuscript. Our responses are provided below, and modifications in the text are shown in red color.

Q1. What is the difference between HVTE and TEHV in the function and concept?

A1. There is an important difference between heart valve tissue engineering (HVTE), which refer to the engineering process, and tissue engineered heart valves (TEHV), which refers to the heart valves! This is the reason we decided to use both the terms and their corresponding abbreviations!

Q2. What are the main factors in positively answering regarding cell differentiation in Decellularized heart valves?

A2. Decellularized heart valves are composed of biological materials that can positively impact cell differentiation and serve as building blocks during the remodeling process.

All these factors are already explained in Acellular scaffolds section, where the authors mentioned: “The advantages of these scaffolds are the removal of all foreign cells and immunogenic compounds and the retaining of their correct anatomical structure and the similar bio-mechanical properties as those of native tissues (like signaling for cell adhesion and induction of cells migration, proliferation and differentiation), that are critical for the long-term functionality of the grafts [20]”.

However, for a better understanding the paragraph was improved as follow:

“The decellularization process consists of removing the cellular material from the ECM of biological tissues, leading to a semi-porous scaffold (remaining ECM), minimizing damage to the original structure and maintaining the same complex geometry of the native tissue. The scaffold obtained contains natural components (collagen, elastin and glycosaminoglycans) that provide clues for cell migration and differentiation, resulting in a constructive remodeling.

Decellularized heart valves have been more clinically relevant than polymeric valves, due to (i) their positively answer regarding cell differentiation (natural components that can positively impact cell differentiation), (ii) the remodeling process, when these serve as building blocks, (iii) the mechanical anisotropy maintaining of the native valves and furthermore, (iv) do not necessitate a complete bio-degradation”.

Q3. Prepare a table and compare the advantages and disadvantages of porous, microsphere, hydrogel, and fibrous scaffolds.

A3. The requested table was prepared and was added to the section 2 “Scaffolds for tissue engineering: general concepts”.

Q4. Page 7/57: this section is not necessary and can be deleted.” The basic principles of cell culture for heart valve tissue engineering commonly involve the following steps”.

A4. The requested paragraph was deleted!

Q5. There are some abbreviations in the tables that require to define first such as in table 1: ECM, table 2: SMA, SMC, MPG, PLDL, …. Please check for all the tables.

A5. The terms SMA, SMC, MPG, PLDL were already explained at the bottom of the mentioned table! However, we carefully checked all the abbreviations from the tables and the missing explanations were added!

Q6. The authors can use the following reference in this manuscript:

Sabbagh, F., Muhamad, I. I., PaLe, N., & Hashim, Z. (2018). Strategies in Improving Properties of Cellulose-Based Hydrogels for Smart Applications. Cellulose-Based Superabsorbent Hydrogels. Springer International Publishing, 887-908.

A6. The mentioned reference is not related to the aim of this manuscript, heart valve tissue engineering. However, the reference was added in the manuscript to the section where cellulose-based hydrogels were discussed.